# Development and crystal structures of a potent second-generation dual degrader of BCL-2 and BCL-xL

Digant Nayak [1,4], Dongwen Lv [1,4], Yaxia Yuan[1], Peiyi Zhang[2], Wanyi Hu[2], Anindita Nayak[1], Eliza A. Ruben[1], Zongyang Lv[1], Patrick Sung [1], Robert Hromas[3], Guangrong Zheng [2]✉, Daohong Zhou[1]✉ & Shaun K. Olsen [1]✉

Overexpression of BCL-xL and BCL-2 play key roles in tumorigenesis and cancer drug resistance. Advances in PROTAC technology facilitated recent development of the first BCL-xL/BCL-2 dual degrader, 753b, a VHL-based degrader with improved potency and reduced toxicity compared to previous small molecule inhibitors. Here, we determine crystal structures of VHL/753b/BCL-xL and VHL/753b/BCL-2 ternary complexes. The two ternary complexes exhibit markedly different architectures that are accompanied by distinct networks of interactions at the VHL/753b-linker/target interfaces. The importance of these interfacial contacts is validated via functional analysis and informed subsequent rational and structure-guided design focused on the 753b linker and BCL-2/BCL-xL warhead. This results in the design of a degrader, WH244, with enhanced potency to degrade BCL-xL/BCL-2 in cells. Using biophysical assays followed by in cell activities, we are able to explain the enhanced target degradation of BCL-xL/BCL-2 in cells. Most PROTACs are empirically designed and lack structural studies, making it challenging to understand their modes of action and specificity. Our work presents a streamlined approach that combines rational design and structure-based insights backed with cell-based studies to develop effective PROTAC-based cancer therapeutics.

The ubiquitin-proteasome system (UPS) is the key cellular machinery responsible for maintaining intracellular protein homeostasis[1,2]. This process is initiated by tagging the target protein with ubiquitin (Ub), which involves sequential interactions and activities of enzymes called E1, E2 and E3, which work together to activate, shuttle and ligate Ub to the target protein, respectively[3,4]. Targeted protein degradation with the use of PROteolysis TArgeting Chimera (PROTACs) that hijack the UPS to specifically degrade a target protein of interest (POI) has attracted great interest in recent years due to the potential of therapeutically modulating proteins that have historically proven difficult to target with conventional small molecule approaches[5,6]. PRO-TACs are heterobifunctional molecules comprising two different ligands connected via an intervening linker[7]. The ligands are specific for an E3 ligase and the target POI, and formation of an E3/PROTAC/POI ternary complex leads to K48-polyUb formation and proteasomal degradation of the POI which otherwise would not be targeted for ubiquitination by the E3 ligase. PROTACs have advantages over traditional small molecule inhibitors as they are event-driven[8] and act sub-

[1]Department of Biochemistry & Structural Biology and Greehey Children's Cancer Research Institute, University of Texas Health Science Center at San Antonio, San Antonio, TX 78229, USA. [2]Department of Medicinal Chemistry, College of Pharmacy, University of Florida, Gainesville, FL 32610, USA. [3]Department of Medicine, University of Texas Health Science Center at San Antonio, San Antonio, TX 78229, USA. [4]These authors contributed equally: Digant Nayak, Dongwen Lv. ✉e-mail: zhengg@cop.ufl.edu; zhoud@uthscsa.edu; olsens@uthscsa.edu

stoichiometrically due to their catalytic mode of action[9,10], have the ability to target "undruggable targets"[11], and can even selectively degrade POI in a tissue/cell specific manner[12]. Interestingly, PROTACs exhibit higher degree of selectivity as compared to the parent ligand molecules due to additional protein-protein interaction (PPI) interfaces generated between the E3 ligase and the noncognate POI during ternary complex formation[13,14]. For these reasons, PROTACs have emerged as a new class of therapeutic molecules to target various pathological proteins and the potential of this approach is demonstrated by the recent surge of molecules already in various stages of clinical development[8,15–17].

Members of the BCL-2 family of proteins are key regulators of cellular apoptosis and include both anti-apoptotic (BCL-2, BCL-xL, MCL-1, etc.) and pro-apoptotic (BAD, BIM, PUMA, BAK, BAX, etc.) proteins[18–20]. The relative abundance of anti- and pro-apoptotic proteins governs cell fate and maintains homeostasis[20]. During oncogenesis, cancer cells evade apoptosis by overexpressing anti-apoptotic proteins which then aid in tumor initiation, progression and development of drug resistance[18]. Hence, much focus has been directed towards development of small molecule inhibitors (SMIs) against BCL-2 family of proteins as a strategy for cancer therapeutics. As a result, a number of SMIs have been discovered, including Navitoclax (ABT263, a dual inhibitor of BCL-xL and BCL-2)[21], Venetoclax (ABT199, selectively inhibits BCL-2)[22], and AMG176[23] (MCL-1 inhibitor), among others[24–29]. Among these inhibitors, Venetoclax is the only FDA-approved anti-tumor drug targeting a BCL-2 family member[30]. However, platelets are dependent on the interaction between BCL-xL and BAK[31], which is disrupted by ABT263. Hence, severe side effects are associated with ABT263, including on-target and dose-limiting thrombocytopenia[32,33]. Moreover, Venetoclax has limited utility for treatment of solid tumors as a large of fraction of the tumors are dependent on BCL-xL but not BCL-2 for survival[34].

To overcome these side effects, we developed PROTAC DT2216 by linking ABT263 with a VHL (E3 ligase) binding ligand[35]. DT2216 has reduced platelet toxicity, as VHL is minimally expressed in platelets and hence BCL-xL is spared in platelets[35]. Surprisingly, DT2216 formed a ternary complex with both BCL-xL and BCL-2 in vitro but it could effectively degrade only BCL-xL but not BCL-2 in cells[35]. Subsequently, we developed the first BCL-xL/BCL-2 dual degrader, 753b, that is more potent than DT2216[36]. However, in the absence of structural studies, the molecular mechanisms of specificity for either of these PROTACs remain unknown.

While important, formation of an E3 ligase-PROTAC-POI ternary complex is not the only criteria for efficient POI degradation. Other factors such as cooperativity, orientation of E3 ligase, and availability of a solvent exposed lysine residues on a POI can also play a crucial role in the effective degradation of POIs via PROTACs[36–40]. Structural studies have been invaluable in driving the discovery and optimization of SMIs against BCL-xL/BCL-2[22,41–44] and VHL-based PROTACs[45–47]; and in this respect, a ternary complex structure could be very informative in improving the efficacy of PROTACs for the POI.

In this study, we set out to determine the molecular basis by which 753b mediates targeted degradation of BCL-xL/BCL-2 and determined crystal structures of both VHL/753b/BCL-xL and VHL/753b/BCL-2 ternary complexes. Interestingly, the two ternary complexes exhibit markedly different architectures that are accompanied by distinct networks of interactions at the VHL/753b linker/target interfaces. We verify the newly created interfacial contacts within the ternary complexes through mutagenesis followed by in vitro ubiquitination assays that are optimized in this study. Subsequently, we embark on a rational and structure-informed design process, harnessing the knowledge gained from our ternary complex structures involving 753b. This endeavor culminates in the development of WH244 PROTAC. Notably, WH244 has demonstrated enhanced potency in degrading BCL-xL/2, as validated through our experiments conducted on human cancer cells. Furthermore, our investigations employing HiBiT, AlphaScreen, AlphaLISA and NanoBRET techniques elucidates the underlying mechanisms responsible for its heightened effectiveness. Lastly, we resolve the crystal structure of the VHL/WH244/BCL-2 ternary complex, offering insights into the molecular mechanisms underpinning its activity. With DT2216 already under clinical trial as the only PROTAC degrader for BCL-xL, our work here lays the foundation for advancing WH244 as a viable candidate for a BCL-xL/BCL-2 dual degrader for use as a cancer therapeutic.

## Results

### Reconstitution of VCB/753b/BCL-xL and VCB/753b/BCL-2 ternary complexes

DT2216 is the only BCL-xL PROTAC currently under clinical investigation. We recently extended our work on DT2216 to generate the first BCL-xL and BCL-2 dual degrader, 753b, through computer modeling-guided rational design (Fig.1a)[36]. 753b is a VHL-based PROTAC and we were interested in elucidating the molecular basis by which 753b induces targeted degradation of BCL-xL/BCL-2 through formation of VHL/753b/BCL-xL and VHL/753b/BCL-2 ternary complexes, respectively. For our structural studies, a complex of VHL, Elongin C, and Elongin B (hereafter referred to as VCB) was used as the E3 substrate receptor subcomplex, as VHL is unstable when expressed alone.

Although there are a growing number of ternary complex structures reported in the Protein Data Bank, crystallizing ternary complexes remains a formidable task. In order to maximize the probability of obtaining diffraction quality crystals, we optimized a biochemical reconstitution strategy for obtaining pure ternary complex. Rather than mixing the proteins (VCB and BCL-xL/BCL-2) and 753b PROTAC at stoichiometric molar ratio and setting up crystallization drops; an additional step was added where the proteins and 753b mixture was further purified using a gel filtration column. As can be seen from the gel filtration profile, the ternary complex was separated from individual subunits and/or binary complexes (VCB/753b, BCL-xL/753b, BCL-2/753b) (Supplementary Fig. 1a–d). Moreover, the retention of ternary complex during the gel filtration served as a proof-of-concept that it is a tightly bound complex. The homogenous ternary complex peak was pooled, concentrated and subjected to crystallization trials. While the strategy we employed to isolate pure ternary complex requires a relatively large amount of PROTAC (~1 mg) to conduct sparse matrix screening and refinement of initial crystal hits, this step was crucial to our crystallization efforts. With respect to applicability to other systems, in a scenario where PROTACs are severely limiting a similar strategy could be employed using size-exclusion chromatography columns with smaller volumes, which could, in principle, reduce the amount of compound required by as much as a factor of 10.

Diffraction quality crystals harboring one complex in the symmetric unit (AU) for each complex appeared in 4–5 days and after extensive optimization, ternary complex structures of VCB/753b/BCL-2 (hereafter referred to as BCL-2[753b]) and VCB/753b/BCL-xL (hereafter referred to as BCL-xL[753b]) were obtained. The final BCL-2[753b] and BCL-xL[753b] structures were resolved at 2.56 Å and 2.94 Å resolution with R/R[free] values of 0.20/0.25 and 0.20/0.24, respectively (Fig.1b *left*, 1c *left*, Supplementary Table 1). In both structures, there is strong and continuous electron density for each component of the complex, and all atoms of the 753b PROTAC could be unambiguously placed (Fig.1b *right*, 1c *right*).

### Overall architecture of the VCB/753b/BCL-2 complex

In the BCL-2[753b] structure, the VHL and BCL-2 ligands engage their respective binding partners in a similar manner as reported for their binary complexes[42,48] (Supplementary Fig. 2a, *left*). Notably, the alkyl chain linker of 753b collapses to bring the VHL ligand very close to α4 helix of BCL-2, in proximity of the BH1 domain (Fig. 2a, g). The ternary complex interface shared between 753b, BCL-2 and VHL can be divided

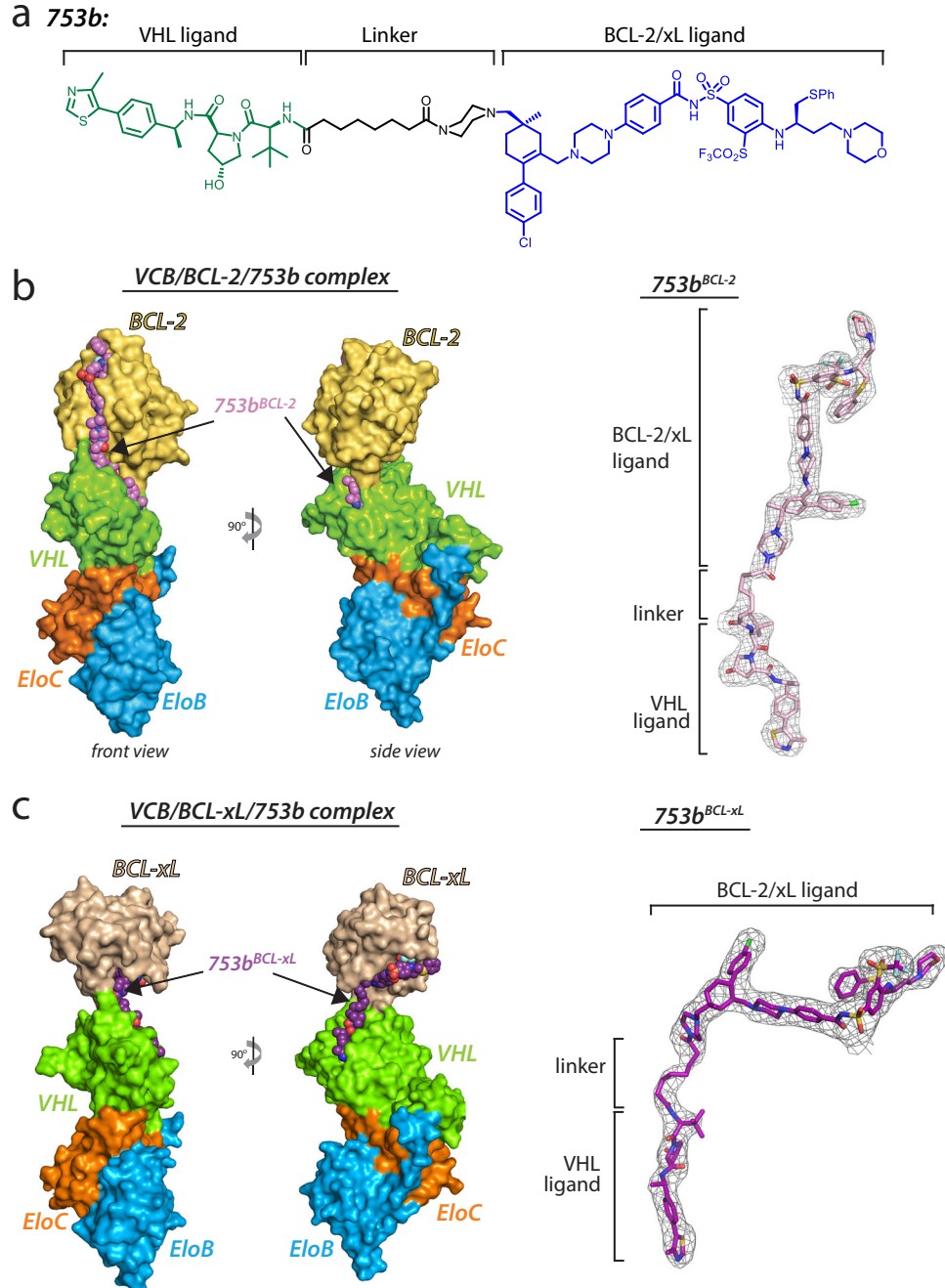

**Fig. 1 | Overall structures of VCB/BCL-2/753b and VCB/BCL-xL/753b ternary complexes. a** Chemical structure of 753b PROTAC. The ligands for VHL and BCL-xL/BCL-2 are colored green and blue, respectively, and the linker is colored black. **b** Front and side views of the VCB/BCL-2/753b structure with proteins presented as surface representation and the 753b PROTAC shown as spheres (left). Composite omit electron density map contoured at 3σ for the 753b PROTAC presented as grey mesh (right). **c** Front and side views of the VCB/BCL-xL/753b structure with proteins presented as surface representation and the 753b PROTAC shown as spheres (left). Composite omit electron density map contoured at 3σ for the 753b PROTAC presented as grey mesh (right).

into three sets of interacting residues. Interestingly, all the BCL-2 contributing residues reside on the α4 helix (Supplementary Fig. 3a). The first set of interactions is mediated by residues that form a hydrophobic patch and constitutes F124, T125 of BCL-2 and Y98, P99 of VHL. This hydrophobic patch is supported by the thiazol-benzyl moiety of 753b (Fig.2a, g). The second set of interactions is shared between G128, T132 of BCL-2, F91 of VHL and the alkyl chain of the linker (Fig. 2a). H115 of VHL participates in hydrogen bonds with the oxygen of amide moiety of linker (closer to VHL ligand) and with the PROTAC. Also, the most prominent de novo PPI observed is the salt bridge

formed between R69 of VHL and E136 of BCL-2 (Fig. 2a, g). R69 (VHL) has been reported to be one of the major contributors for PPIs with the POI in many studies[45,46,49]. Although, as noted earlier the overall structures of both VHL and BCL-2 in BCL-2[753b] remain unchanged as compared to their binary complexes, on closer inspection we did observe modest structural changes in residues involved in intermolecular interactions. Both R69 of VHL and E136 of BCL-2, for instance, adopted alternate rotamers that promote a salt bridge between them and helps stabilize the ternary complex (Supplementary Fig. 2a, *right*).

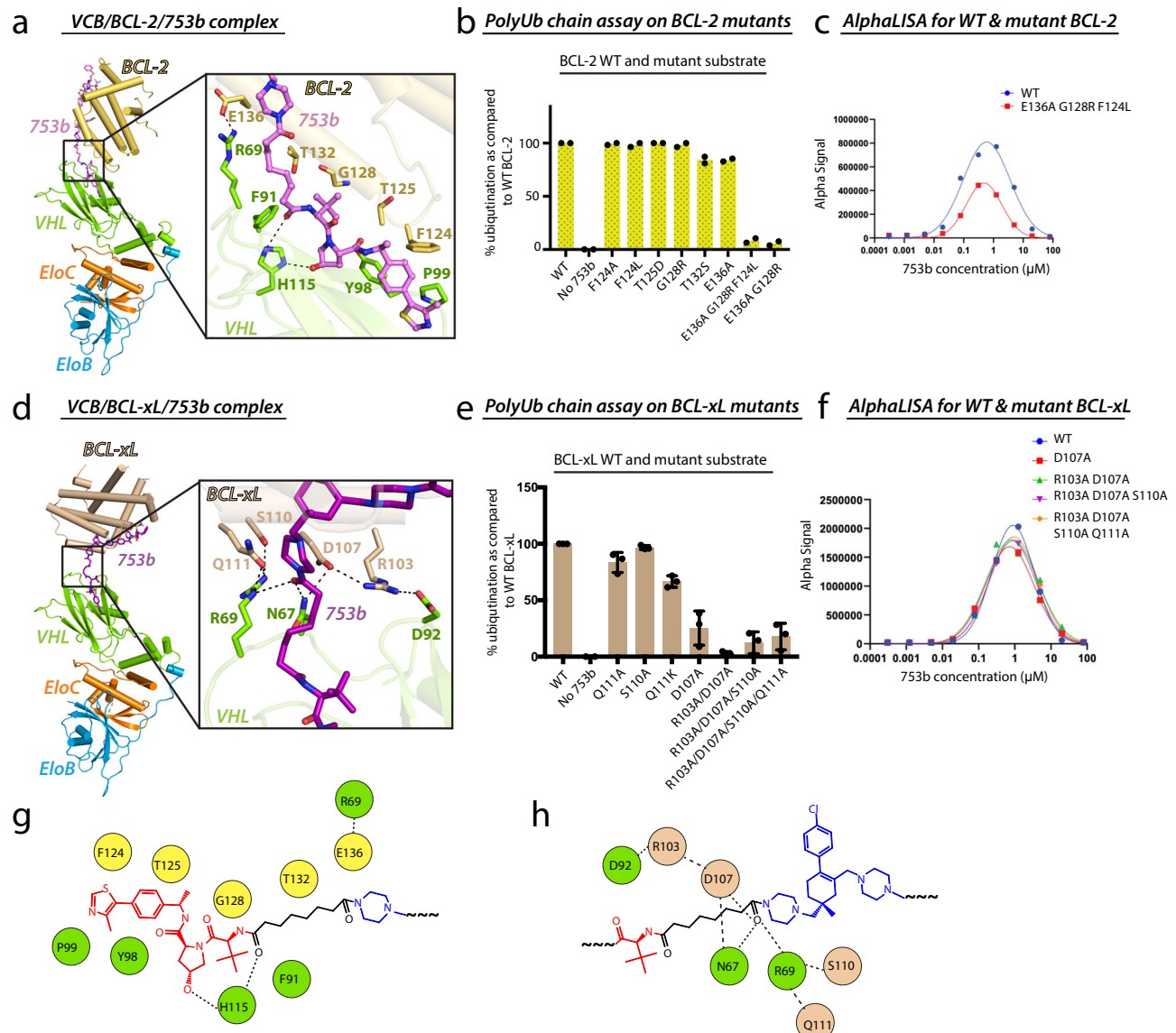

**Fig. 2 | Distinct networks of interactions in the VCB/753b/BCL-2 and VCB/753b/BCL-xL ternary complexes. a** Overview of the VCB/BCL-2/753b structure shown as cartoon representation (left) alongside a magnified view of the interaction interface (right). Residues involved in contacts between 753b linker, BCL-2 and VHL are shown as sticks. Oxygen atoms are colored red, nitrogen atoms blue, and dashed lines indicate hydrogen bonds. **b** PolyUb chain formation assay using the indicated mutants of BCL-2 as identified based on the interaction networks presented in (**a**) (right). Data represented as bar graph with mean of 2 technical repeats, displayed as percentage value of the WT value. **c** Assessment of ternary complex formation by WT and mutant BCL-2 with VCB and 753b in a cell-free condition using AlphaLISA assay. Data represent the mean of a single experiment with 2 technical replicates. **d** The VCB/BCL-xL/753b structure is presented as in (**a**). **e** PolyUb chain formation assay using the indicated mutants of BCL-xL based on the interaction networks presented in (**d**) (right). Data represented as bar graph with mean ± SD, of 3 technical repeats, displayed as percentage value of the WT value. **f** Assessment of ternary complex formation by WT and mutant BCL-xL with VCB and 753b in a cell-free condition using AlphaLISA assay. Data represent the mean of a single experiment with 2 technical replicates. **g, h** Schematic representation highlighting the interaction network between VHL, 753b linker, and BCL-2 (**a**) and BCL-xL (**d**), respectively. Source data are provided as Source data file.

We next set out to assess the functional importance of the interfacial residues identified in our BCL-2[753b] structure. Thus, we generated mutants designed to disrupt the PPIs and protein-753b linker interactions followed by a biochemical assay where we reconstituted all the active proteins in vitro to catalyze the polyubiquitin chain formation on BCL-2 in the presence of 753b, which would be the readout for the assay (Supplementary Fig. 4a, b). Consistent with the structure, E136A/G128R and E136A/G128R/F124L mutations in BCL-2 that are predicted to disrupt the ternary complex exhibited a complete lack of polyubiquitin chain formation (Fig. 2b, Supplementary Figs. 4c, 5a; Lane 9 and 10). F124 was a key residue supporting the hydrophobic patch created by Y98 (VHL) and thiazol-benzyl moiety of 753b (Fig. 2a, g). Based on our structure, the E136A/

G128R/F124L mutant should have destabilized the complex as it affects the entire stretch of region on the α4 helix that contacts VHL and 753b. Importantly, all the mutants generated were stable and well-folded as compared to wild-type BCL-2 as assessed by thermal stability assays (Supplementary Fig. 6). Moreover, in order to further verify whether the loss of activity is due to the mutation of interfacial residues on BCL-2 and not because of lack of binary binding to 753b or lack of ternary complex formation, we performed AlphaScreen and AlphaLISA assays. There was a marginal increment in the binary Kd value of the BCL-2 triple mutant (E136A/G128R/F124L) as compared to WT BCL-2 (Supplementary Fig. 7f–h). Moreover, compared to WT BCL-2, the triple mutant formed a weaker ternary complex as evidenced by reduction in Alpha signal (Fig. 2c). Collectively, these

studies demonstrate the importance of interfacial contacts as observed in the crystal structure.

## Overall architecture of the VCB/753b/BCL-xL complex

BCL-xL and BCL-2 share a modest sequence identity (41%) and structural similarity (overall RMSD of ~1.66 Å)[50] (Supplementary Fig. 8). Hence, it was surprising to observe that, the relative orientation of BCL-xL in BCL-xL[753b] was drastically different as compared to BCL-2 in BCL-2[753b] (Supplementary Fig. 3b), despite being co-crystallized with the same PROTAC molecule harboring the same POI ligand i.e. ABT263 (Supplementary Fig. 2a, b, *left*). Superposition of the VHL subunits from the BCL-xL[753b] and BCL-2[753b] structures reveals that BCL-xL and BCL-2 are nearly 180° relative to each other along with a ~14 Å translation due to the altered path of the PROTAC (Supplementary Fig. 3b). Unlike the BCL-2[753b] complex, BCL-xL does not interact with the VHL ligand. Moreover, the majority of PPIs and protein-753b linker interactions were mediated by electrostatic interactions of residues residing on the α3 helix of BCL-xL, centered on one end of the 753b linker (Fig. 2d, h, Supplementary Fig. 3a). The oxygen of amide moiety of the linker (closer to ABT263 ligand) is positioned such that it makes an extensive network of hydrogen bonds with D107 (BCL-xL), R69 (VHL) and N67 (VHL) (Fig. 2d, h). As a result, this network is further extended as D107 (BCL-xL) simultaneously contacts both R103 (BCL-xL) and N67 (VHL). R103 (BCL-xL) also acts as a bridge by contacting both D107 (BCL-xL) and D92 (VHL) (Fig. 2d, h). Additionally, the interface appears to be stabilized by R69 (VHL), which makes multiple hydrogen bonds with carboxamide of Q111 (BCL-xL) and hydroxyl of S110 (BCL-xL) (Fig. 2d). Although, the overall structures of BCL-xL and VHL in the BCL-xL[753b] ternary complex remains the same as compared to the respective binary structures (VHL/ligand and BCL-xL/ligand), there appear to be minor variations in the residues responsible for PPIs and protein-ligand interactions. For BCL-xL, there was distinct reorientation of R103 and S110 in order to generate neo interactions with VHL and stabilize the complex (Supplementary Fig. 2b). However, for VHL, no major difference was observed (Supplementary Fig. 2b).

We next assessed the importance of the ternary complex interactions observed in our BCL-xL[753b] structure by subjecting BCL-xL mutants designed to disrupt PPIs and protein-753b linker interactions to the polyubiquitin chain formation assays described above (Supplementary Fig. 4d, e). Although the loss of polyubiquitination was not significant for S110A and Q111A BCL-xL mutants (Fig. 2e and Supplementary Figs. 4f, 5b; lane 3 and 4), there was a significant loss of activity for D107A (Fig. 2e and Supplementary Figs. 4f, 5b; lane 6), consistent with D107 being a central residue at the ternary complex interface where it coordinates multiple interactions between BCL-xL, VHL and the linker moiety. Interestingly, while Q111A had little impact on BCL-xL polyubiquitination, a charge swap mutant Q111K was more defective as compared to Q111A (Fig. 2e). Moreover, higher order mutants were more effective in abrogating the polyubiquitin chain formation on BCL-xL (Fig. 2e and Supplementary Fig. 5b; lane 7, 8 and 9). Importantly, all of the mutants generated were stable and well-folded compared to wild-type BCL-xL, as assessed by thermal stability assays (Supplementary Fig. 9). Furthermore, to ascertain that the decline in activity is indeed attributed to the mutation of interfacial residues on BCL-xL, we conducted AlphaScreen assay (for measuring binary binding) and AlphaLISA assay (for measuring ternary complex formation) as discussed above. Notably, all the defective mutants had more or less similar binary affinity to 753b (Supplementary Fig. 7a–e, h) and they exhibited minor reduction in ternary complex formation as compared to WT BCL-xL (Fig. 2f). Taken together, these in vitro experiments validate the importance of PPIs and protein-ligand interactions identified in the VCB/753b/BCL-xL ternary complex structure.

## Rational and structure-guided design of WH244, a more potent dual degrader of BCL-xL/BCL-2

During the drug discovery process, availability of a ligand-bound structure can guide the design of subsequent generations of compounds with enhanced potency and efficacy. DT2216 harbored an aliphatic chain as linker (5-mer alkyl chain), which was connected to the VHL and BCL-xL/BCL-2 binder via amide bonds. Although, DT2216 could form tight ternary complexes with both BCL-2 and BCL-xL in vitro, it could degrade only BCL-xL in cells[35]. Among the possible explanations for this discrepancy are that full-length BCL-xL and BCL-2 were expressed in cells for the nanoBRET assay, whereas recombinant-transmembrane-domain-deleted proteins were used for the AlphaLISA experiment and/or that in cells the surface of BCL-2 engaging DT2216/VHL is sterically occluded by other factors, such as protein binding partners. With the aid of computer modeling and linker length optimization, 753b was generated, where a different link-out position on BCL-xL/BCL-2 ligand (ABT263) was chosen, with a 6-mer alkyl chain as the linker[36]. Based on our two ternary complex structures with 753b, we observed that the 753b linker was instrumental in mediating the neo-PPIs. Hence, acknowledging the fact that linker geometry could play an important role in defining the ternary complex arrangement, we developed a PROTAC, WH244, in which we replaced the flexible alkyl chain of 753b with a more rigid 1,4-dimethylpiperazine (Fig. 3a). Notably, the WH244 linker is of similar length as that of 753b which seemed optimal based on the ternary structures. Further, the WH244 linker was designed with the same chemical composition (amide bonds) at its junctions with the VHL and BCL-xL/2 ligands as in 753b, since one of the amide oxygens makes key contacts in both the BCL-2[753b] and BCL-xL[753b] ternary complex structures. We reasoned that a rigid linker would maintain the overall tertiary arrangement of the complex as 753b to efficiently degrade BCL-xL and BCL-2[36]. Furthermore, another improvement was made wherein the morpholine group in the BCL-xL/BCL-2 warhead was replaced with a one-carbon bridged morpholine (1R, 4R −2 oxa-5-azabicyclo 2.2.1 heptane) (Fig. 3a), with the intention of decreasing lipophilicity[51].

A series of experiments were performed to assess the efficacy of WH244. Interestingly, it exhibited higher potency as compared to 753b with a $DC_{50}$ of 0.6 nM (BCL-xL) and 7.4 nM (BCL-2) (Fig. 3b). These $DC_{50}$ values were ~6.2 and 6.8-fold better than of 753b for BCL-xL ($DC_{50}$ of 3.7 nM) and BCL-2 ($DC_{50}$ of 50 nM), respectively. Also, WH244-mediated proteolysis of BCL-xL and BCL-2 was time-dependent with BCL-xL degrading faster than BCL-2 (Fig. 3c). It was interesting to observe that the degradation of BCL-xL and BCL-2 could be seen as early as 1 h after treatment and nearly complete after 4 h (BCL-xL) and 6 h (BCL-2), respectively. Notably, protein levels for both BCL-xL and BCL-2 recovered to pretreatment levels within 24–48 h after removal of WH244, indicating that the degradation observed is reversible after the removal of the degrader due to new protein synthesis (Fig. 3d). Moreover, we found that pretreatment of the cells with ABT263 or VHL ligand (VHL-L), completely blocked the degradation of BCL-xL and BCL-2 with WH244 (Fig. 3e). In order to confirm that the proteolysis of BCL-xL/BCL-2 is UPS mediated, we treated the cells with MLN4924 and MG132, which inhibit NEDD8-activating enzyme and proteasome, respectively. This treatment abrogated WH244-mediated degradation of BCL-xL and BCL-2 (Fig. 3f). Next, we wanted to evaluate the anti-tumor efficacy of WH244 and tested it in Jurkat cells, which are dependent on both BCL-xL and BCL-2 for survival. Based on the viability experiment, WH244 exhibited enhanced potency and outperformed 753b, DT2216, and ABT263 in killing Jurkat cells (Fig. 3g). Encouraged by our previous study on 753b[36], we performed co-immunoprecipitation (co-IP) assay to determine whether K17 on BCL-2 still remains the sole lysine for polyubiquitin chain formation. The results showed that under WH244 treatment, K17 was still the primary lysine for ubiquitination (Fig. 3h, *left* Lanes 2 and 6). Surprisingly, unlike 753b, K22 showed weak ubiquitination activity and appeared to

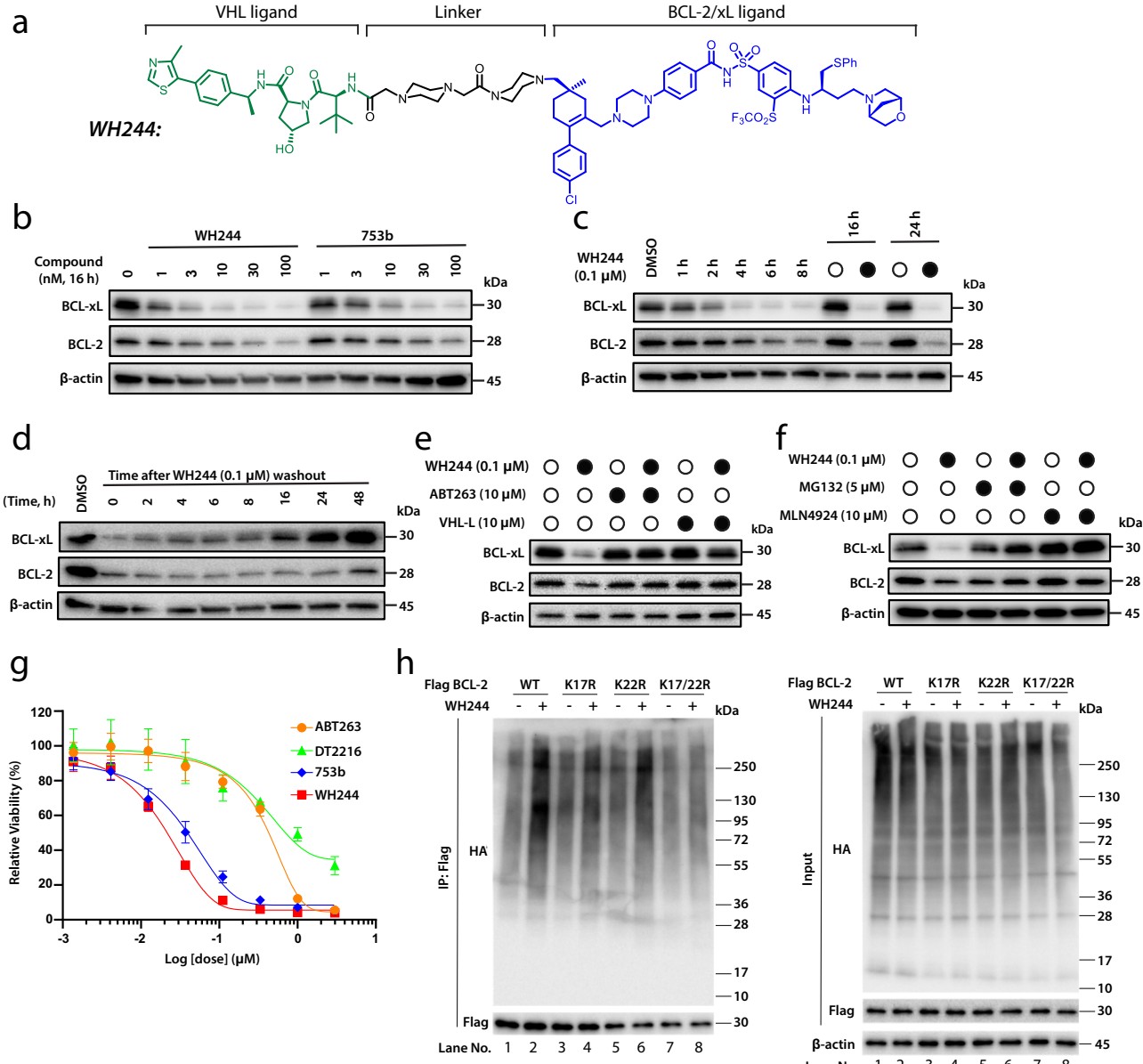

**Fig. 3 | WH244 degrades both BCL-xL/BCL-2 and is more potent than 753b.**
**a** Chemical structure of WH244 PROTAC. **b** The potency of WH244 and 753b in degrading BCL-xL/BCL-2 was evaluated in Jurkat cells by immunoblots after the cells were treated with different concentrations of WH244 and 753b for 16 h. **c** The time course of WH244-mediated BCL-xL/BCL-2 degradation was evaluated in Jurkat cells by immunoblots after the cells were treated with 0.1 μM WH244 for various time points as indicated. **d** To test the lasting effect of WH244, cells were treated with 0.1 μM WH244 for 16 h before replating them on a new plate with fresh medium followed by immunoblotting of BCL-xL/BCL-2 at indicated time points. **e** Pretreatment with ABT263 or VHL Ligand (VHL-L) for two hours followed by six-hour treatment with WH244 blocks the BCL-xL/BCL-2 degradation induced by WH244 in Jurkat cells. **f** Inhibition of proteasomes and CRL[VHL] neddylation using MG132 and MLN4924, respectively, blocks BCL-xL/BCL-2 degradation induced by

WH244 in Jurkat cells. **g** Cell proliferation assay in Jurkat cells was determined after they were incubated with increasing concentrations of ABT263, DT2216, 753b and WH244 for 72 h. The data are presented as mean ± s.d. from three replicate cell cultures in a representative experiment for Jurkat cells. **h** *Left*, WH244 induces polyubiquitination of WT, K22R BCL-2 and to some extent K17R but not K17/22R BCL-2 in 293T cells. 293T cells were co-transfected as indicated with Flag-BCL−2 WT or mutants and HA-tagged ubiquitin (HA-Ub) plasmids. After 36 h, cells were pre-treated with MG132 (10 μM) for 2 h and then treated with or without WH244 (1 μM) for 5 h. The Flag-tagged proteins were immunoprecipitated (IP) and immuno-blotted with HA or Flag antibody. *Right*, The input blot for this assay. Data are a representative of two independent experiments. Source data are provided as Source Data file.

play a contributory role (Fig. 3h, *left* Lanes 2, 4 and 8). Collectively, these studies confirm WH244-mediated degradation of BCL-xL and BCL-2 in a VHL- and UPS-dependent manner, demonstrates that WH244 has better antitumor activity as compared to previously generated PROTACs for BCL-xL/BCL-2[35,36,49] and illustrates the lysine specificity for WH244-mediated polyubiquitination of BCL-2.

To further evaluate the degradation efficiencies of BCL-xL and BCL-2 induced by WH244 and 753b, we generated endogenous HiBiT

knock-in BCL-xL and HiBiT knock-in BCL-2 HeLa cells through CRISPR-Cas9 gene editing (Supplementary Fig. 10a). As demonstrated in our previous report[36], HeLa cell is a BCL-xL/2-independent cancer cell line, which can be used to test PROTACs (against BCL-xL/2) with minimized confounding effect of apoptosis on protein degradation. We compared the degradation of BCL-xL and BCL-2 induced by WH244 and 753b using the HiBiT knock-in HeLa cell lines. The results showed that WH244 is more potent than 753b for inducing the degradation of both

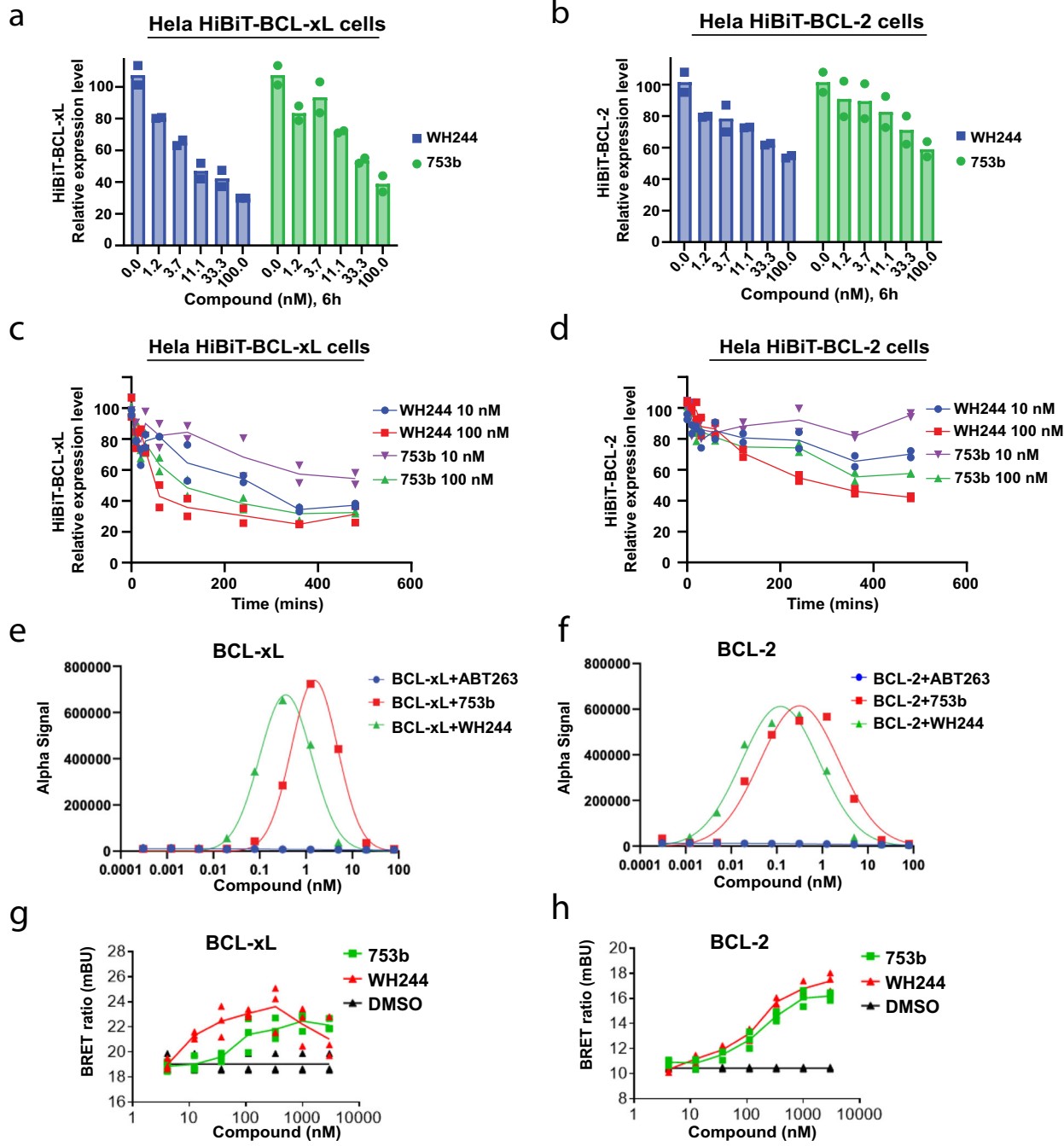

**Fig. 4 | Evaluating the degradation efficiency and ternary complex formation ability of BCL-xL/2 induced by WH244 and 753b, respectively. a, b** HiBiT degradation assay to measure the HiBiT-BCL-xL (**a**) and HiBiT-BCL-2 (**b**) degradation after 6 h treatment with different doses of WH244 or 753b. Data are represented as mean of two biological replicates. **c, d** HiBiT degradation assay to measure the time-dependent HiBiT-BCL-xL (**c**) and HiBiT-BCL-2 (**d**) degradation with two different doses of WH244 or 753b. Data are represented as mean of two biological replicates. **e, f** The formation of VCB/PROTAC/BCL-xL (**e**) and VCB/ PROTAC/BCL-2 (**f**) ternary complexes in a cell-free condition was determined by AlphaLISA assay. Data represent the mean of a single experiment with 2 technical replicates. **g, h** Cellular ternary complex formation induced by 753b and WH244. 293T cells were transiently transfected with HiBiT-BCL-xL, LgBiT and HaloTag-VHL (**g**) or HiBiT-BCL-2, LgBiT and HaloTag-VHL (**h**) and then treated with a serial dilution of 753b or WH244 for 4 h. Cells were pretreated with proteasome inhibitor MG132 for 2 h to block the degradation of BCL-xL/2. Data are expressed as mean of three biological replicates. Source data are provided as Source data file.

BCL-xL and BCL-2, respectively (Fig. 4a, b). We further performed the time course assay in the edited cells and confirmed that WH244 induced faster degradation of BCL-xL and BCL-2 than 753b (Fig. 4c, d). Compared with BCL-2, BCL-xL is more susceptible to be degraded by our BCL-xL/2 PROTACs. At the concentration of 100 nM, WH244 and 753b degraded 50% of BCL-xL in ~45 min and ~84 min, respectively. At the concentration of 10 nM, WH244 required ~219 min to degrade 50% of BCL-xL but 753b was unable to achieve 50% degradation within

480 min (Fig. 4c). In terms of BCL-2, 100 nM WH244 degraded 50% of BCL-2 in about 250 min but 753b failed to achieve 50% degradation in 480 min (Fig. 4d).

To further assess the cellular selectivity of 753b and WH244 for BCL-xL/BCL-2 proteins, multiplexed tandem mass tag (TMT) labeling mass spectrometry proteomic experiments were performed to monitor protein levels in an unbiased fashion. Jurkat cells were treated in triplicate with DMSO, 100 nM 753b or 100 nM WH244 for 8 h

(Supplementary Fig. 10b). The TMT results identified approximately 6000 proteins, none of which exhibited significant down-regulation following PROTAC treatment, indicating that our BCL-xL and BCL-2 dual PROTACs have limited off-target effects. With that said, it is worth noting that BCL-xL and BCL-2 were surprisingly absent from these results. To assess BCL-xL and BCL-2 degradation, we utilized western blotting with the same samples, and successfully verified the degradation of both BCL-xL and BCL-2 proteins induced by 753b and WH244, respectively (Supplementary Fig. 10c). We speculate that these apparently contradictory results may be attributable to some of the limitations of TMT technology, where low TMT-labeling efficiency for some proteins and peptides, can lead to inability to detect these proteins by TMT-based proteomics analysis. This limitation has been observed in several other reported PROTAC studies[52,53].

Given the somewhat inconclusive nature of the TMT results, we made additional efforts to characterize the on-target and off-target effects of our BCL-xL and BCL-2 dual PROTACs using a different proteomics analysis technique, i.e. the Data-Independent Acquisition (DIA) proteomics approach. This approach can avoid the limitation of TMT-labeling deficiency for some proteins/peptides. Specifically, we conducted DIA proteomics profiling using HeLa cells because these cells are primarily dependent on MCL-1 for survival and thus resistant to apoptosis after BCL-xL and BCL-2 degradation. To our delight, we were able to identify over 7200 proteins, including both BCL-xL and BCL-2 (Supplementary Fig. 10d), using this method. Notably, both 753b and WH244 demonstrated the ability to degrade BCL-xL and BCL-2, with WH244 exhibiting superior efficacy in the degradation of both proteins (Supplementary Fig. 10e). Additionally, DIA proteomics revealed perturbations in a few other proteins, which remains to be determined whether these proteins represent downstream targets regulated by BCL-xL/2 or real off-targets of these PROTACs. Overall, based on the two different proteomic analyses, we were able to confirm that WH244 and 753b can effectively degrade BCL-xL and BCL-2 with limited off-target effects. In addition, the DIA proteomics analysis also confirmed that WH244 is more potent than 753b in degrading BCL-xL and exhibits slightly greater efficacy in degrading BCL-2.

**Basis for enhanced WH244 degrader activity**

We further performed a series of experiments to explain the enhanced degrading capability of WH244. We conducted an in vitro ternary complex formation assay via AlphaLISA and observed that both BCL-xL and BCL-2 formed a tighter complex with WH244 as compared to 753b (Fig. 4e, f). Moreover, the cellular NanoBRET assay demonstrated that WH244 initiated ternary formation at a significantly lower compound concentration than 753b for BCL-xL, indicating a more robust complex (Fig. 4g). However, for BCL-2, WH244 was modestly better than 753b in inducing ternary complex formation in cell (Fig. 4h). Moreover, based on the binary binding assays, the Kd for WT BCL-xL and WT BCL-2 against 753b was ~28 nM and ~23.4 nM, respectively, whereas for WH244 it was ~6.14 nM and ~3.1 nM, respectively (Supplementary Fig. 11a, b). These experiments suggested that the increased effectiveness of WH244 may be attributed to refinements in the linker and the BCL-xL/BCL-2 warhead, resulting in more robust binary binding and, consequently, a tighter ternary complex formation when compared to 753b.

To further delve into the molecular mechanism of WH244 and motivated by our success in obtaining two ternary complex crystal structures with 753b, we next attempted to co-crystallize WH244-mediated ternary complexes in order to determine the basis for the enhanced potency of WH244. Although we were unable to obtain crystals of VCB/WH244/BCL-xL, we successfully obtained diffraction quality crystals for VCB/WH244/BCL-2 (hereafter referred to as BCL-2$^{WH244}$) using the in-solution ternary complex purification strategy (Supplementary Fig. 1e, 1f) described earlier. The BCL-2$^{WH244}$ structure was refined to 2.98 Å resolution with R/R$_{free}$ value of 0.23/0.29 and one

complex in the symmetric unit (AU) (Fig. 5a, Supplementary Table. 1). Moreover, there was strong and continuous electron density for each component of the complex including the WH244 PROTAC, which as will be described in more detail below, adopts a distinct conformation at the VHL/BCL-2 interface, relative to the 753b-bound structures (Fig. 5b).

WH244 in the BCL-2$^{WH244}$ ternary complex structure adopts what resembles a "horseshoe" conformation with the linker forming the base of the horseshoe and the VHL and BCL-2 ligands forming the sides. This organization results in the α3 helix of BCL-2 being positioned in proximity to VHL. A cluster of salt bridges and hydrogen bonds are observed at the ternary complex interface centered around the carbonyl oxygen of the amide moiety of the WH244 linker proximal to the VHL ligand (Fig. 5c). Here, N67 of VHL participates in hydrogen bonds with both the carbonyl oxygen of the WH244 linker and the side chain of R110 of BCL-2, which also forms a salt bridge with D92 of VHL (Fig. 5c). Further reinforcing this interface, R69 of VHL forms a salt bridge with E114 of BCL-2 which also interacts with the amino-moiety of piperazine linker of WH244. Additionally, Q118 of BCL-2 contacts the amino-moiety of piperazine that is part of ABT263 ligand (Fig. 5c). To assess the importance of the interfacial residues identified in the BCL-2$^{WH244}$ structure, we employed the polyubiquitin chain formation assay described above for interrogation of 753-bound ternary complex structures. (Supplementary Fig. 4g, h). A single point mutation of R110A on BCL-2 exhibits severely diminished poly-ubiquitination in the assay (Fig. 5d and Supplementary Fig. 5c, Lane no. 3), demonstrating the importance of the contiguous tripartite network of hydrogen bonds extending from the WH244 linker to VHL via BCL-2 (Fig. 5c, d). While E114A retained minimal activity as compared to R110A, double mutants R110A/E114A and R110A/E114K of BCL-2 residues at this tripartite interface completely abolished BCL-2 polyubiquitination except E114A/Q118A, which retained limited levels of activity relative to wild type (Fig. 5d, Supplementary Fig. 2i, Supplementary Fig. 5c; Lane 4, 5, 7 and 8). Furthermore, our AlphaLISA assay indicated that functionally defective mutants of BCL-2 formed a considerably weaker ternary complex (Fig. 5e), without a notable decrease in their binary binding affinity to WH244 (Supplementary Fig. 12a–i). This further affirms the significance of interfacial contacts, as observed in the crystal structure.

Comparison of the BCL-2$^{WH244}$ ternary complex to the BCL-xL$^{753b}$ and BCL-2$^{753b}$ complexes reveals that the orientation of BCL-2 in BCL-2$^{WH244}$ is completely changed relative to that of BCL-2 in the BCL-2$^{753b}$ structure and instead resembles that of BCL-xL in the BCL-xL$^{753b}$ structure (Fig. 6a, b). This conformational difference is accompanied by distinct sets of BCL-2 residues being involved in interactions at the BCL-2$^{WH244}$ ternary complex interfaces compared to those in the BCL-2$^{753b}$ structure. In the BCL-2$^{WH244}$ structure, BCL-2 residues involved in interactions at the ternary complex interface reside within the α3 helix, whereas in the BCL-2$^{753b}$ structure, the α4 helix of BCL-2 is involved in most of the interactions at the ternary complex interface (Supplementary Fig. 13). Interestingly, this new set of residues on BCL-2 that are involved in ternary complex interactions in the BCL-2$^{WH244}$ structure are conserved in BCL-xL, where they also mediate ternary complex interactions in the BCL-xL$^{753b}$ structure (Supplementary Fig. 13). Moreover, this PPI/protein-linker interface of both BCL-xL and BCL-2 share a sequence similarity of 70.5%. Also, the majority of contacts to the PROTAC are electrostatic in the BCL-2$^{WH244}$ structure, whereas they are largely hydrophobic in the BCL-2$^{753b}$ structure. Furthermore, intriguing findings emerged concerning the bridged morpholine and piperazine linker within WH244. In contrast to the morpholine group in 753b, the bridged variant in WH244 established additional interactions with BCL-2 (Fig. 6c). Upon aligning the BCL-2 from both the ternary structures, a slight conformational shift in the bridged morpholine of WH244 was observed, facilitating the establishment of more interactions (Fig. 6c). Moreover, E114 on BCL-2 engaged in electrostatic

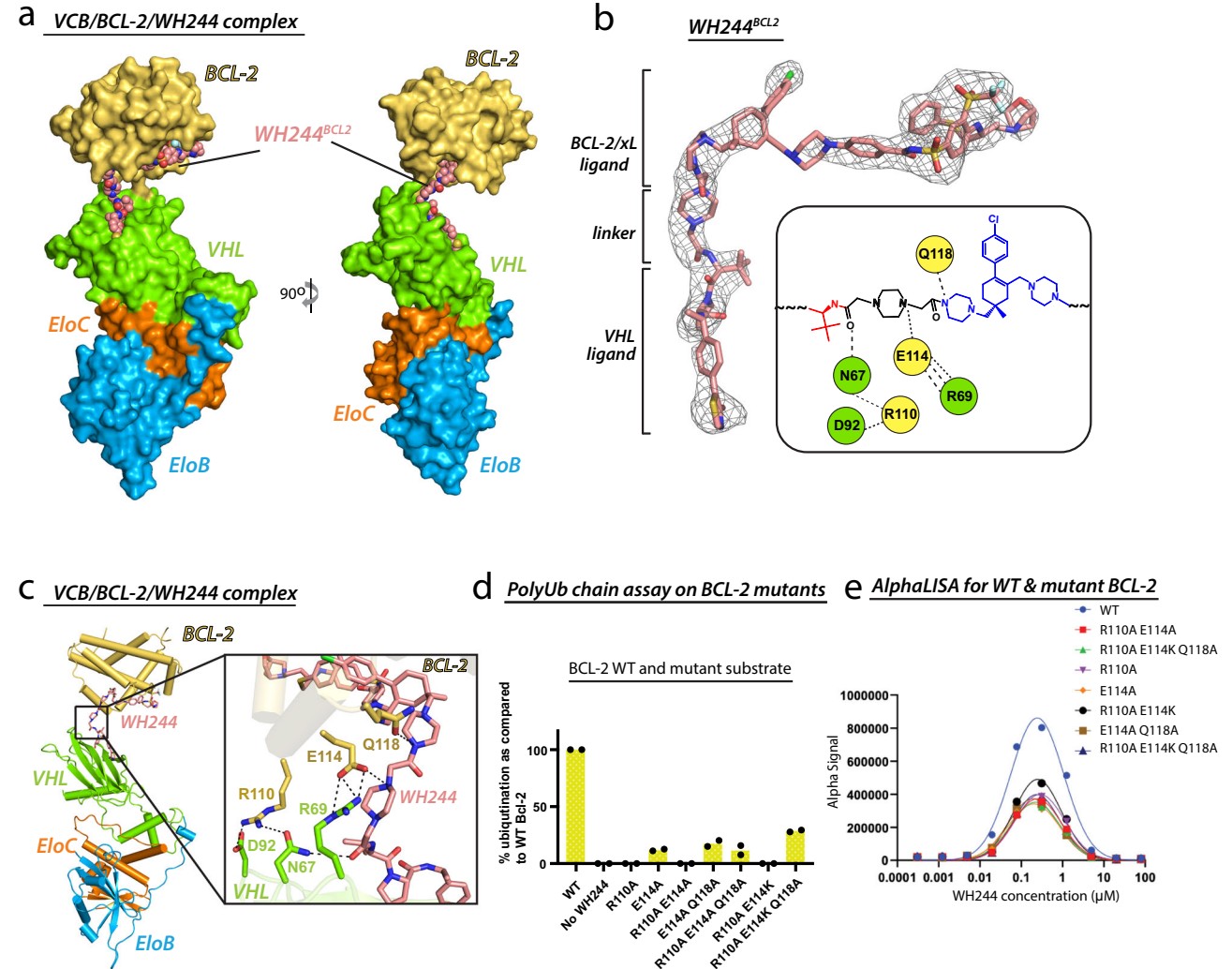

**Fig. 5 | Overall architecture and structure-function analysis of the VCB/WH244/BCL-2 ternary complex. a** Front and side views of the VCB/BCL-2/WH244 structure with proteins presented as surface representation and the WH244 PROTAC shown as spheres. **b** Composite omit electron density map contoured at 3σ for the WH244 PROTAC presented as grey mesh (left) and a cartoon schematic representation of the interaction network between WH244 linker and residues from BCL-2 and VHL (right). **c** Overview of the VCB/BCL-2/WH244 structure shown as cartoon representation (left) alongside a magnified view of the interaction interface (right). Residues involved in contacts between WH244 linker, BCL-2 and VHL are shown as sticks. Oxygen atoms are colored red, nitrogen atoms blue, and dashed lines indicate hydrogen bonds. **d** PolyUb chain formation assay using the indicated mutants of BCL-2 as identified based on the contacts seen in (c) (left). Data represented as bar graph with mean of 2 technical repeats, displayed as percentage value of the WT value. **e** Assessment of ternary complex formation by WT and mutant BCL-2 with VCB and WH244 in a cell-free condition using AlphaLISA assay. Data represent the mean of a single experiment with 2 technical replicates. Source data are provided as Source data file.

contacts with the piperazine linker of WH244 (Fig. 6d). In summary, these findings collectively demonstrate that the enhanced efficacy of WH244 can be ascribed to the integration of a polar 1,4-dimethylpiperazine linker and the utilization of a bridged morpholine in the BCL-xL/BCL-2 warhead. This integration resulted in a more robust binary complex and, consequently, a significantly stronger ternary complex of VCB/WH244/BCL-2.

## Discussion

In this study, we sought to understand the molecular mechanism of 753b, a dual degrader for BCL-xL/BCL-2 through structural studies and whether this insight can be exploited to effectively develop a more potent degrader. Our work was motivated by a previous study, that identified 753b as the only PROTAC molecule that can degrade both BCL-xL and BCL-2 in cells. However its mode of engagement with either of the proteins in context to VHL (E3 ligase) remained unknown[36]. We solved the crystal structures of VCB/753b/BCL-xL and

VCB/753b/BCL-2, respectively, which revealed a constellation of interactions mediated by VHL (E3 ligase), PROTAC-linker (753b) and POIs (BCL-xL/BCL-2). To our knowledge, this is the first reported structure of a PROTAC mediated ternary complex with BCL-2. Strikingly, the relative orientations of BCL-xL and BCL-2 were very different with respect to the aligned VHL (E3 ligase), despite the fact that both the proteins share a high structural similarity[50]. Moreover, in addition to the molecular insights achieved for 753b engagement, the ternary structures also laid the groundwork for the development of a more efficient PROTAC, WH244, for which we also determined the ternary complex structure with VCB and BCL-2. Altogether, this work provides a good example for rational design supplemented with structural insights for PROTAC development.

Almost all the reported studies on PROTACs have utilized various biophysical techniques like isothermal titration calorimetry (ITC), AlphaLISA, fluorescence polarimetry (FP), etc. to assess different aspects of the ternary complexation. In the present study, we wanted

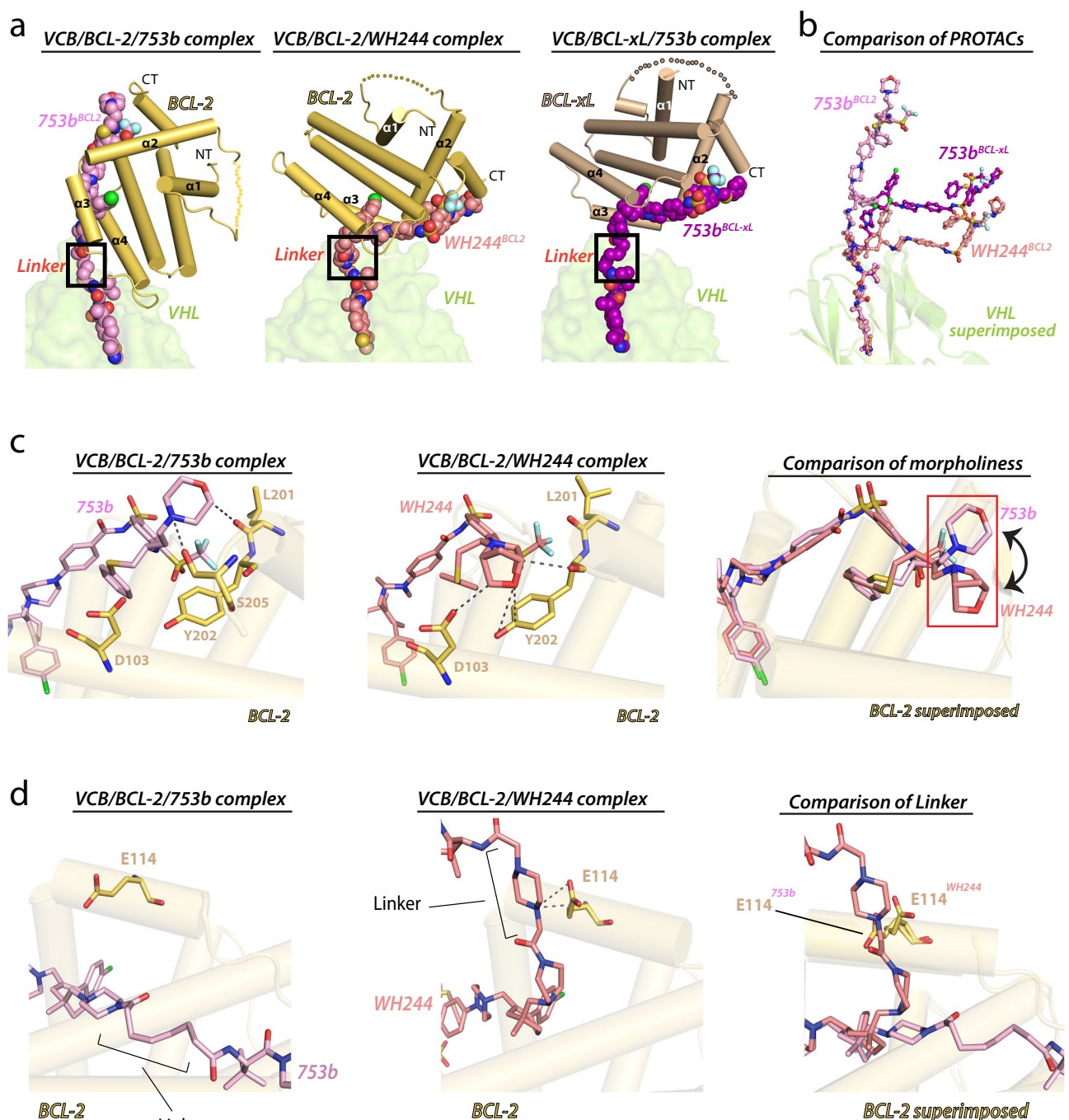

**Fig. 6 | Differences in linker and morpholine groups between 753b and WH244 leads to tight binding and enhanced activity. a** The indicated ternary complex structures are shown with proteins presented as cartoon representations and the PROTACs presented as spheres. Disordered loops are connected by semi-transparent spheres. **b** Differences in the orientation of PROTACs in the VCB/target/PROTAC ternary complexes. VHL from the VCB/BCL-2/753b, VCB/BCL-xL/753b, VCB/BCL-2/WH244 complexes were superimposed. VHL is shown as semi-transparent surface, PROTACs are shown as sticks. BCL-2/xL are not shown for clarity. **c** Comparing the interactions mediated by morpholine group in VCB/BCL-2/753b (left) and bridged morpholine group in VCB/BCL-2/WH244 (center).

Right, differences in the orientation of morpholine and bridged morpholine, respectively when BCL-2 from VCB/BCL-2/753b and VCB/BCL-2/WH244 complexes were superimposed. BCL-2 shown as transparent cartoon. Residues involved in contacts are shown as sticks. Oxygen atoms are colored red, nitrogen atoms blue, and dashed lines indicate hydrogen bonds. **d** Glu114 (BCL-2) in VCB/BCL-2/753b (left) complex does not make any contact with the 753b, whereas in same orientation it forms two hydrogen bonds with piperazine of WH244 of VCB/BCL-2/WH244 (center) complex. Right, BCL-2 from both the complexes were superimposed to show relative position of Glu114 and the linker.

to assess the interfacial contacts identified based on our crystal structure and we developed an in vitro ubiquitination assay. This method requires limited material, is label-free, occurs in-solution, has an easy readout through immunoblotting of ubiquitinated POI, and most importantly, mimics the physiological environment as it involves

native proteins. The results from our study show the relevance of this bioassay, which was crucial in assessing the importance of key interacting residues on BCL-xL/BCL-2 through mutagenesis. Nevertheless, it's worth noting that our bioassay may not serve as a direct substitute for well-established techniques such as ITC, FP and AlphaLISA. These

established methods offer comprehensive and quantitative data regarding ternary complex formation, providing valuable insights into the interactions and binding affinities involved. While our bioassay has its merits, it may not fully encompass the level of quantitative detail and precision that these conventional techniques can deliver. However, our bioassay could be added to the suite of various biophysical assays used in PROTAC discovery which could provide complementary information. Furthermore, it is essential to note that the mutual orientations of VHL, PROTAC, and BCL-xL/BCL-2, as depicted in this study, may not precisely mirror those observed in living cells. This discrepancy arises from the limitations imposed by crystal packing in our ternary structures (Supplementary Fig. 14). Nevertheless, the deductions drawn from our comprehensive array of biochemical and biophysical assessments strongly indicate that our ternary structures have indeed captured functionally pertinent intermediates with 753b and WH244, respectively.

This study further bolsters the power of rational design with structural insights to enhance PROTAC efficacy. As mentioned before, a tight co-operative ternary complex is the first step to targeted degradation of a POI. Factors like linker chemistry/geometry and its link-out location on the E3 ligase/ POI ligand also play a very significant role in PROTAC enhancement[36–38]. In our previous work with 753b, we had also optimized its linker length to 6-mer alkyl chain to achieve maximum potency[36]. Therefore, utilizing this information in conjunction with our structural data, we made a significant modification by substituting the linker in 753b with a 1,4-dimethylpiperazine moiety. This choice was guided by the linker's similar length to the 6-mer alkyl chain and to increase rigidity[54]. Additionally, the presence of an ionizable amino group in 1,4-dimethylpiperazine could potentially facilitate electrostatic interactions. Simultaneously, we pursued a rational improvement of 753b by introducing a bridging carbon atom in the morpholine group of BCL-xL/BCL-2 warhead (Fig. 3a). This design rationale was influenced by the enhanced physicochemical properties often associated with bicyclic compounds[51,55]. Our biophysical findings underscored that these modifications resulted in a more robust binding of WH244 to BCL-xL/BCL-2, consequently leading to the formation of a more stable ternary complex (Fig. 4e, f). Furthermore, we successfully elucidated the mechanism of WH244 by resolving the VCB/WH244/BCL-2 structure, shedding light on the additional interactions facilitated by the bridged morpholine and piperazine in WH244. These interactions fortified the binary complex formed with BCL-xL/BCL-2, ultimately contributing to the heightened potency of WH244. In essence, our structural insights played a pivotal role in unveiling the mechanism underpinning WH244's efficacy.

Resistance to apoptosis is a hallmark for cancer and overexpression of anti-apoptotic proteins like BCL-xL/BCL-2 are partly responsible for it[18,56,57]. Although DT2216 is the only PROTAC molecule undergoing clinical trials for degrading BCL-xL, it is less effective in treating solid tumors and some specific forms of leukemia that are co-dependent on both BCL-xL and BCL-2 for survival, when administered alone and needs to be supplemented with ABT199 for better efficacy[11]. Moreover, SMIs for both BCL-xL and BCL-2 suffer from normal tissue toxicity, particularly thrombocytopenia[32]. Studies have shown that leukemia patients undergoing venetoclax treatment regime experience relapse due to appearance of venetoclax resistance. The tumor cell develops this resistance in part by overexpressing anti-apoptotic proteins MCL-1 and BCL-XL[58]. For such situations, PROTAC technology could be the answer as they function in a catalytic manner with minimal dose requirement (as compared to SMIs) and most importantly their effect is event- and not occupancy-driven which minimizes resistance development[8,59,60]. Moreover, unlike inhibitors, PROTACs degrade the POI and in a way, "completes the job". Although, 753b was efficient in degrading both BCL-xL and BCL-2 and inducing apoptosis in AML cells[36], based on our cellular studies WH244 was more potent in killing Jurkat cancer cells (also dependent on BCL-xL/BCL-2) as

compared to other SMIs, DT2216 and 753b (Fig. 3g). Our live-cell HiBiT degradation assay further confirmed the better potency of WH244 as compared to 753b (Fig. 4a, b). Lastly, the WH244-based ternary structure provides a platform for further development of a safe and effective therapeutic targeting BCL-xL/BCL-2 dependent malignancies.

## Methods
### Cloning
Elongin B (full-length) and Elongin C (aa 17-112) co-expression pACYC-Duet1 plasmid was purchased from Addgene (Addgene plasmid # 110274). cDNA of VHL (aa 54-213) was cloned into pET28a with thrombin cleavable N-terminal 6xHis tag using Gibson Assembly. For BCL-xL (aa 2-212) and BCL-2 (aa 1-207), the corresponding DNA fragments were codon-optimized for *Escherichia coli* expression by Gene Universal (USA) and cloned into NdeI/NotI and BamHI/XhoI sites of modified vector pSMT3.4 and pET29a, with an N-terminal ULP1-cleavable SMT3 tag and TEV cleavable N-terminal 6xHis tag, respectively. All point mutations were introduced using PCR-based site-directed mutagenesis. All the plasmids were transformed into BL21 (DE3) codon plus *E. coli* strain for protein expression. Single point mutations were generated using the primer pairs described in Supplementary Table 2. Double or triple mutants were ordered from by Gene Universal (USA).

### Protein expression and purification
BCL-xL (aa 2-212) and BCL-2 (aa 1-207) transformed cells were grown in LB broth at 37 °C with shaking until the optical density at 600 nm reached 1.5. Isopropyl-β-D-thiogalactopyranoside (0.3 mM) was added to induce protein expression overnight at 18 °C. VHL (aa 54-213) and ElonginBC were co-transformed (VCB) and induced as indicated above. Cell pellet was resuspended in lysis buffer 50 mM tris-Cl (pH 8.0), 300 mM NaCl, 5% Glycerol, 5 mM β-mercaptoethanol, 10 mM imidazole and lysed using sonication. The lysate was cleared by centrifugation at $35,000 \times g$ for 30 min at 4 °C and applied to Ni-NTA agarose (Qiagen) beads, and the protein was eluted in buffer 50 mM tris-Cl (pH 8.0), 200 mM NaCl, 5 mM β-mercaptoethanol and 300 mM imidazole. The affinity tag was retained for VCB complex, while 6xHis tag on BCL-2 and SMT3 tag on BCL-xL were cleaved by adding TEV protease (at 1:100 w/w) and SUMO ULP protease (at 1:1000 w/w) and incubating at 4 °C overnight, respectively. For BCL-xL, the sample was passed through fresh Ni-NTA agarose (Qiagen) beads to remove the cleaved tag, uncleaved BCL-xL and ULP protease and further purified using gel filtration (Superdex 75; GE Healthcare). For BCL-2, the sample was purified using gel filtration (Superdex 75; GE Healthcare) followed by anion-exchange (MonoQ 10/100; GE Healthcare) chromatography. For VCB complex, after elution from Ni-NTA beads, it was directly subjected to gel filtration (Superdex 200; GE Healthcare) followed by anion-exchange (MonoQ 10/100; GE Healthcare) chromatography. All the proteins were concentrated to 4–8 mg/ml, aliquoted, and snap-frozen in liquid nitrogen. Similar purification protocol was used for the all the mutants used in the study.

For polyubiquitination assay, all the necessary proteins were recombinantly produced. S. *pombe* Uba1 (E1), human versions of E2s (UbcH5b and Cdc34) and Ubiquitin was produced in bacteria as described earlier[61,62]. Briefly, following tag cleavage, Uba1 was purified via using gel filtration (Superdex 200; GE Healthcare) followed by anion-exchange (MonoQ 10/100; GE Healthcare) chromatography. E2s and Ubiquitin were gel filtrated using Superdex 75; GE Healthcare followed by rebinding to Ni-NTA agarose (Qiagen) beads to remove any uncleaved proteins. Both Wild-type Cullin2 and Rbx1 were cloned in pFastbac-Dual vector with GST tag on Cullin2 and expressed in *Trichoplusia ni* High-Five (BTI-TN-5B1-4) insect cells, followed by affinity purification with Glutathione beads, overnight PreScission cleavage and size-exclusion chromatography. Purification of NEDD8, Ubc12, APPBP1–UBA3 and neddylation of cullin2-Rbx1, were performed as previously described[63]. Briefly, NEDD8 E1 was purified as

GST-tagged protein with the tag on APPBP1 and purified via Superdex 200; GE Healthcare upon tag cleavage. Both Ubc9 and NEDD8 were purified via 6X-His tag on N-terminal followed by Superdex 75; GE Healthcare after tag cleavage from Ubc9. For neddylation reaction, 0.1 μM Uba1, 2 μM Ubc9, 5 μM Cul2/Rbx1 and 10 μM NEDD8 were mixed in a buffer composition of 20 mM HEPES pH 7.5, 100 mM NaCl, 0.5 mM DTT, 10 mM $MgCl_2$ and 2 mM ATP at room temperature for 20–30 min. The neddylated Cul2/Rbx1 was further purified by passing the reaction mixture through Ni-NTA agarose (Qiagen) beads. Upon imidazole elution, the neddylated Cul2/Rbx1 was passed though Superdex 200; GE Healthcare.

### Crystallization and data collection

VCB, BCL-xL and 753b were mixed in 1:1:3 molar ratio and incubated for 30 min at room temperature. The ternary complex was purified by gel filtration (Superdex 200; GE Healthcare) and concentrated to 14-16 mg/ml. VCB/BCL-xL/753b crystals were grown by mixing 0.3-μl protein and 0.3-μl well solution containing 0.1 M Sodium Cacodylate pH 6.0, 0.2 M Sodium chloride and 8% PEG 8000 on a 96-well sitting plate at 18 °C. The crystals appeared in 4–5 days and were further improved using Additive screen (Hampton). The best crystals appeared in 0.1 M Sodium Cacodylate pH 5.5, 0.2 M Sodium chloride, 6-8% PEG 8000 and 4% tert-Butanediol. Crystals were cryo-protected by well solution supplemented with 25% (v/v) ethylene glycol and snap-frozen in liquid nitrogen.

The ternary complex of VCB/BCL-2/753b (at approximately 10–12 mg/ml) was prepared and crystalized using the similar approach as for VCB/BCL-xL/753b. Crystals appeared in well solution containing 0.1 M Tris-Cl pH 8.5, 20% PEG 8000, 0.2 M $MgCl_2$ within 2–4 days. After various optimizations, the best crystals appeared in 0.1 M Tris-Cl pH 8.7, 18% PEG 8000, 0.2 M $MgCl_2$ and 3% 1,6-Hexanediol and cryo-protected by well solution supplemented with 25% (v/v) ethylene glycol and snap-frozen.

Using similar protocol, the ternary complex of VCB/BCL-2/WH244 at 11–14 mg/ml was prepared and taken for crystallization trials. Crystals appeared in well solution containing 0.1 M Sodium Cacodylate pH 6.5, 0.3 M sodium malonate dibasic monohydrate, 8% γ-PGA (PGA screen, Molecular Dimensions) within 2–4 days. After various optimizations, the best crystals appeared in 0.1 M Sodium Cacodylate pH 6.7, 0.2 M sodium malonate dibasic monohydrate, 8.5% γ-PGA (PGA screen, Molecular Dimensions) and cryo-protected by well solution supplemented with 25% (v/v) ethylene glycol and snap-frozen.

### Structure determination and refinement

A complete data set was collected from the VCB/BCL-xL/753b, VCB/BCL-2/753b and VCB/BCL-2/WH244 crystals to 2.94, 2.56 and 2.98 Å resolution, respectively at the Advanced Photon Source, NE-CAT beamline 24-IDE at a wavelength of 0.979 Å. Datasets were indexed, integrated, and scaled using XDS[64].

For VCB/BCL-xL/753b, the crystal belonged to space group $P2_12_12_1$ with unit cell dimensions $a = 47.62$ Å, $b = 118.88$ Å, $c = 170.11$ Å, and $β = 90°$. There was one ternary complex per asymmetric unit. The structure was solved by molecular replacement using the program PHASER[65]. The search models used were VCB structure (PDB: 1VCB) and BCL-xL (PDB: 1R2D). Model and restraints for 753b was generated using Phenix.Elbow[66]. Model of VCB/BCL-xL/753b was subjected to iterative rounds of refinement and rebuilding using PHENIX[67] and COOT[68]. The final model for VCB/BCL-xL/753b complex has $R/R_{free}$ values of 0.198/0.242. The structure has an excellent geometry as assessed using Molprobity with following Ramachandran parameters: favored (96.6%), allowed (3.4%), and outliers (0.0%).

For VCB/BCL-2/753b, the crystal belongs to space group $P12_11$ with unit cell dimensions $a = 47.48$ Å, $b = 94.59$ Å, $c = 81.18$ Å and $α = γ = 90°$, $β = 97.45°$. There was one ternary complex per asymmetric unit. The structure was solved by molecular replacement using the program PHASER. The search models used were VCB structure (PDB: 1VCB) and BCL-2 (PDB: 5JSN). Structure with 753b was refined as described above. The final model for VCB/BCL-2/753b complex was refined to $R/R_{free}$ values of 0.195/0.2490 via iterative rounds of refinement and rebuilding using PHENIX[67] and COOT[68]. The final Ramachandran stats are favored (95.1%), allowed (4.7%), and outliers (0.2%) suggesting nice geometry.

For VCB/BCL-2/WH244, the crystal belongs to space group $P2_12_12_1$ with unit cell dimensions $a = 47.30$ Å, $b = 102.60$ Å, $c = 167.40$ Å and $β = 90°$. There was one ternary complex per asymmetric unit. The structure was solved by molecular replacement using the program PHASER. The search models used were VCB structure (PDB: 1VCB) and BCL-2 (PDB: 5JSN). Model and restraints for WH244 was generated using Phenix.Elbow[66]. Structure with WH244 was refined as described above. The final model for VCB/BCL-2/WH244 complex has $R/R_{free}$ values of 0.234/0.289. The final Ramachandran stats are favored (93.0%), allowed (6.6%), and outliers (0.4%) suggesting nice geometry.

### Cell lines and cell culture

Hela (Cat. No. CCL-2) and Jurkat (Cat. No. TIB-152) cells were recently purchased from American Type Culture Collection (ATCC, Manassas, VA, USA). Hela cells were cultured in complete Dulbecco's modified Eagle medium (DMEM, Cat. No. 12430054, Thermo Fisher Scientific) supplemented with 10% (vol/vol) heat-inactivated fetal bovine serum (FBS, Cat. No. S11150H, Atlanta Biologicals, Flowery Branch, GA, USA), 100 U/ml penicillin and 100 μg/ml streptomycin (penicillin–streptomycin, Cat. No. 15140122, Thermo Fisher Scientific). Jurkat cells were cultured in RPMI 1640 medium (Cat. No. 22400–089, Thermo Fisher Scientific) supplemented with 10% FBS, 100 U/ml penicillin and 100 μg/ml streptomycin. All cells were maintained in a humidified incubator at 37 °C and 5% $CO_2$. Human platelet-rich plasma (PRP) was purchased from Zenbio (Durham, NC) (cat. no. SER-PRP-SDS).

### Immunoblotting

Cells were collected, washed once with ice-cold phosphate-buffered saline, pH 7.2 (PBS, cat. no. 20012027; Thermo Fisher Scientific) and lysed in RIPA lysis buffer (Cat. No. BP-115DG, Boston Bio Products, Ashland, MA, USA) supplemented with protease and phosphatase inhibitor cocktails (Cat. No. PPC1010, Sigma-Aldrich, St. Louis, MO, USA) through sonication. The samples were centrifuged at $15,000 \times g$ for 5 min and the supernatants were transferred into a new tube. The protein concentration in the supernatants was determined using the Pierce BCA protein Assay kit (cat. no. 23225, Thermo Fisher Scientific). The protein concentration was normalized, and the samples were reduced in 4× Laemmli's SDS-sample buffer (cat. no. BP-110R, Boston Bio Products) and denatured at 95 °C for 5 min. An equal amount of protein samples (20–40 μg per lane) were resolved using precast 4–20% Tris-glycine gels (Mini-PROTEAN TGX, cat. no. 456–1094, Bio-Rad), and resolved proteins were transferred onto 0.2-μm pore size polyvinylidene difluoride (PVDF) blotting membranes (cat. no. LC2002, Thermo Fisher Scientific) using mini trans-blot electrophoretic transfer cell (Bio-Rad). The membranes were blocked with non-fat dry milk (5% wt/vol) in 1×Tris- buffered saline-Tween-20 (TBST) for 1 h at room temperature, and were subsequently probed with primary antibodies at a pre-determined optimal concentration in non-fat dry milk (5% wt/vol in TBST) overnight at 4 °C. The membranes were washed three times (15 min each) in TBST and then incubated with horse radish peroxidase (HRP)-conjugated secondary antibodies for 1 h at room temperature. Following sufficient washing with TBST, the membranes were exposed with ECL Western Blotting Substrate (Bio-Rad), and the signal was detected using the ChemiDoc MP Imaging System (Bio-Rad) and quantified using the ImageJ (v1.53a) software from NIH. Antibodies were purchased from Cell Signaling Technologies (CST) and the dilutions are as follows: BCL-xL (Cat. No. 2762S,

1:1000) and BCL-2 (Cat. No. 2870S, 1:1000). β-actin antibody was purchased from MP Biomedicals (Cat. No. 8691001, 1:20,000).

For detection of polyubiquitin chain formation on BCL-xL and BCL-2 in in vitro assay, following antibodies were used: BCL-xL (Cat. No. sc-8392, 1:1000) and BCL-2 (Cat. No. sc-7832, 1:1000). The remaining protocol remains same as above.

## Binary binding affinities

To test the binary/ternary binding affinities of compounds to BCL-2/BCL-xL, AlphaScreen displacement binding assay was performed to test the capabilities of testing compounds to displace a Biotin-tagged Bim (Biotin-MRPEIWIAQELRRIGDEFNA) or Bad (Biotin-LWAAQRY-GRELRRMSDEFEGSFKGL) from BCL-2 or BCL-xL for BCL-2 and BCL-xL binding affinities, respectively. All reagents were diluted in assay buffer of 25 mM HEPES, pH 7.5, 150 mM NaCl, 0.1% BSA, and 0.005% tween 20 prior incubation. To a 96-well PCR plate (VWR, Cat. No. 82006-636) was added 10 μL Biotin-tagged Bim (15 nM for WT or 50 nM for mutants) or Bad (15 nM for WT or 50 nM for mutants), 10 μL series diluted compounds solution (2 or 3 or 4× dilution), and 10 μL self-made 6×His tagged BCL-2 WT or mutants (25 nM WT or 50 nM mutants) or 10 μL self-made BCL-xL WT and mutants (10 nM WT or 50 nM mutants). After incubating at room temperature for 30 min, 5 μL of 160 μg/mL of anti-6×His acceptor beads (PerkinElmer, Cat. No. AL128M) was added and the mixture was incubated at room temperature for 15 min. 5 μL of 160 μg/mL of streptavidin donor beads (PerkinElmer, 6760002) was then added under subdued light and the mixture was incubated at dark for 30 min before being transferred to two adjacent wells (17 μL each) of 384-well white OptiPlate (PerkinElmer, Cat. No. 6008280). The luminescence signal was detected on a Biotek's Synergy Neo2 multimode plate reader installed with an AphaScreen filter cube. The Ki values of each compounds were calculated by fitting data into the "Binding-competitive, One site-Fit Ki" function in GraphPad Prism (9.2.0) based on concentrations of Biotin−peptide used and experimentally determined $K_d$ values of each 6×His protein &&&and Biotin−Peptide pair.

## AlphaLISA ternary complex assay

Assays were performed at room temperature. All reagents were diluted in assay buffer of 25 mM HEPES, pH 7.5, 150 mM NaCl, 0.1% BSA, and 0.005% tween 20. To a 96-well PCR plate (VWR, Cat. No. 82006-636) were added 10 μL self-made 6×His tagged BCL-xL (100 nM) or BCL-2 (150 nM), 10 μL series diluted compounds solution (4× dilution) and 10 μL of self-made GST-VCB (100 nM). After incubating at room temperature for 15 min, 5 μL of alpha glutathione-donor beads (160 μg/mL, PerkinElmer, Cat. No. 6765300) and 5 μL of anti-6×His acceptor beads (160 μg/mL, PerkinElmer, Cat. No. AL128M) to each well and incubated for additional 45 min in the dark before being transferred to two adjacent wells (17 μL each) of 384-well white OptiPlate (PerkinElmer, Cat. No. 6008280). The luminescence signal was detected on a Biotek's Synergy Neo2 multimode plate reader installed with an AphaScreen filter cube. Intensity values were plotted in Graphpad Prism (9.2.0) with PROTAC concentration values represented on a log10 scale.

## Viability assays

For the viability assays in Jurkat cells, the cells in complete cell culture medium were seeded in 96-well plates (100 μl per well) at the optimized densities (50,000 cells). Compound treatments were prepared in complete cell culture medium and 100 μl of 2× treatment-containing medium were added to each well. Complete cell culture medium without treatment was added in control wells, and wells containing medium without cells served as background. The outer wells of the 96-well plate were not used for treatment and were filled with 250 μl of PBS to reduce evaporation of medium from inner wells. Each compound and/or combination was tested at nine different concentrations with three replicates. The cell viability was measured after 72 h by

tetrazolium-based MTS assay. MTS reagent (2 mg mL⁻¹ stock; cat. no. G1111, Promega) was freshly supplemented with phenazine methosulfate (PMS, 0.92 mg mL⁻¹ stock, cat. no. P9625, Sigma-Aldrich) in 20:1 ratio, and 20 μl of this mixture was added to each control and treatment well. The cells were incubated for 4 h at 37 °C and 5% CO$_2$, and then the absorbance was recorded at 490 nm using Biotek's Synergy Neo2 multimode plate reader (Biotek). The average absorbance value of background wells was subtracted from absorbance value of each control and treatment well, and percent cell viability ((At/A0) × 100) was determined in each treatment well, where At is the absorbance value of the treatment well and A0 is the average absorbance value of control wells after background subtraction. The data were expressed as average percentage cell viability and fitted in non-linear regression curves using GraphPad Prism 7 (GraphPad Software, La Jolla, CA, USA).

## Generation of endogenous HiBiT knock-in BCL-xL/2 HeLa cells through CRISPR-Cas9 gene editing

The Alt-R CRISPR RNA (crRNA):Alt-R trans-activating crRNA (tracrRNA) duplexes for targeting BCL-xL or BCL-2 were prepared by mixing 0.5 nmol crRNA and 0.5 nmol tracrRNA in 25 μL of Nuclease-Free Duplex Buffer (Integrated DNA Technologies, IDT) by incubation at 95 °C for 5 min and cooling to RT. Single-stranded ultramer DNA oligonucleotides (ssODN) (IDT) were used as the homology directed repair (HDR) donor templates. The sequences of gRNAs and ssODN templates are listed in Supplementary Table 2. Ribonucleoprotein (RNP) complexes with recombinant *Streptococcus pyogenes* Cas9 protein (IDT) and crRNA:tracrRNA duplex were assembled by incubating 100 pmol of Cas9 and 120 pmol of gRNA for 10−20 min at room temperature. HeLa ($2 × 10^5$) cells were resuspended in 10 μL of Neon™ transfection system buffer R from the Neon™ transfection system 10 mL kit (Thermo Fisher), and the RNP complex along with 100 pmol ssODN template were electroporated into HeLa cells with the Neon™ transfection system (Thermo Fisher). After electroporation, HeLa cells were transferred to a 12-well plate for culturing. Edited pooled cells were analyzed for HiBiT insertion by assaying for luminescence using Nano-Glo® HiBiT lytic detection system (Cat #N3040, Promega) in a Biotek's Synergy Neo2 multimode plate reader (Biotek) 48−72 h after electroporation and single clones were selected by serial dilution.

## HiBiT degradation assay

HeLa HiBiT-BCL-xL and HeLa HiBiT-BCL-2 cells ($2 × 10^4$/well in 80 μL DMEM) were seeded into white 96-well tissue culture plates and incubated overnight at 37 °C, 5% CO$_2$. For dose curve testing, the following day, serial diluted compounds were added into the medium and then plates were incubated at 37 °C, 5% CO$_2$, for 6 h. For time course testing, in the following day, compounds with indicated concentrations were added into the medium at different time points. After treatment, the luminescence signals were measured using Nano-Glo® HiBiT lytic detection system (Cat #N3040, Promega) in a Biotek's Synergy Neo2 multimode plate reader (Biotek).

## Thermal shift assay

Thermostability of BCL-xL/BCL-2 and mutants were detected using the GloMelt thermal shift protein stabilization kit (Biotium) according to the manufacturer instruction. 10 μL of the protein (2.0 mg/mL) was mixed with 89.5 μL of assay buffer (10 mM HEPES, 150 mM NaCl, pH 7.4) and 0.5 μL of GloMelt thermoshift dye. Each mixture was then dispensed into a 96-well qPCR plate (3 × 20 μL). The denaturation of the protein was then measured with a Quantstudio 3 thermal cycler (Thermo Fisher) and during a melt curve ranging from 25−95 °C by 0.02 °C/s steps. The derivatives of the fluorescent signal changes were plotted against temperature, and the temperature with minimal derivative was recorded as the melting temperature (Tm).

## Polyubiquitin chain formation assay

WT BCL-xL/BCL-2 (5 µM) were ubiquitinated in the presence of E1 *Sp*Uba1 (0.1 µM), E2, UbcH5c (1 µM) and Cdc34 (1 µM), ubiquitin (75 µM), VHL-Elongin C-Elongin B complex (0.35 µM), neddylated Cullin2-Rbx1 complex (0.35 µM) and 753b or WH244 (10 µM) at 37 °C in a buffer of 50 mM HEPES, pH 7.5, 10 mM $MgCl_2$, 100 mM NaCl, 2 mM ATP, and 0.5 mM DTT. The reaction mixture was incubated for indicated time points (Supplementary Fig. 4a, b, d, e, g, h) and was terminated by the addition of 1× reducing SDS-PAGE loading buffer. The polyUb chain formation was assessed using immunoblotting as described earlier. For polyUb chain formation on mutant proteins used in this study, the above protocol was used with a fixed time-point of 1 h.

## TMT-based proteomics

**Sample preparation and LC-MS/MS analysis.** Total protein from cell pellets was reduced, alkylated, and purified by chloroform/methanol extraction prior to digestion with sequencing grade modified porcine trypsin (Cat. No. V5111, Promega). Tryptic peptides were labeled using tandem mass tag isobaric labeling reagents (Cat. No. A34808, Thermo Fisher) following the manufacturer's instructions and combined into one 11-plex sample group. The labeled peptide multiplex was separated into 46 fractions on a 100 × 1.0 mm Acquity BEH C18 column (Waters) using an UltiMate 3000 UHPLC system (Thermo) with a 50 min gradient from 99:1 to 60:40 buffer A:B (Buffer A: 0.1% formic acid, 0.5% acetonitrile; Buffer B: 0.1% formic acid, 99.9% acetonitrile. Both buffers adjusted to pH 10 with ammonium hydroxide for offline separation) ratio under basic pH conditions, and then consolidated into 18 super-fractions. Each super-fraction was then further separated by reverse phase XSelect CSH C18 2.5 µm resin (Waters) on an in-line 150 × 0.075 mm column using an UltiMate 3000 RSLCnano system (Thermo). Peptides were eluted using a 75 min gradient from 98:2 to 60:40 buffer A:B ratio. Eluted peptides were ionized by electrospray (2.4 kV) followed by mass spectrometric analysis on an Orbitrap Eclipse Tribrid mass spectrometer (Thermo) using multi-notch MS3 parameters. MS data were acquired using the FTMS analyzer in top-speed profile mode at a resolution of 120,000 over a range of 375–1500 m/z. Following CID activation with normalized collision energy of 35.0, MS/MS data were acquired using the ion trap analyzer in centroid mode and normal mass range. Using synchronous precursor selection, up to 10 MS/MS precursors were selected for HCD activation with normalized collision energy of 65.0, followed by acquisition of MS3 reporter ion data using the FTMS analyzer in profile mode at a resolution of 50,000 over a range of 100–500 m/z.

**Data processing and analysis.** Proteins were identified and MS3 reporter ions quantified using MaxQuant (version 2.2.0.0; Max Planck Institute) against the Homo sapiens UniprotKB database (March 2023) with a parent ion tolerance of 3 ppm, a fragment ion tolerance of 0.5 Da, and a reporter ion tolerance of 0.003 Da. Scaffold Q+S (Proteome Software) was used to verify MS/MS based peptide and protein identifications protein identifications were accepted if they could be established with less than 1.0% false discovery and contained at least 2 identified peptides; protein probabilities were assigned by the Protein Prophet algorithm[69] and to perform reporter ion-based statistical analysis. Protein TMT MS3 reporter ion intensity values were assessed for quality using ProteiNorm for a systematic evaluation of normalization methods[70]. Cyclic loess normalization was utilized since it had the highest intragroup correlation and the lowest variance amongst the samples. Statistical analysis was performed using Linear Models for Microarray Data (limma) with empirical Bayes (eBayes) smoothing to the standard errors[71]. Proteins with *p*-value < 0.05 and fold change >2 were considered to be significant.

## DIA proteomics

**Sample preparation and LC-MS/MS analysis.** Cells were lysed in buffer containing 5% SDS/50 mM triethylammonium bicarbonate (TEAB) in the presence of protease and phosphatase inhibitors (Halt; Thermo Scientific) and nuclease (Pierce™ Universal Nuclease for Cell Lysis; Thermo Scientific). Aliquots corresponding to 100 µg protein (EZQ™ Protein Quantitation Kit; Thermo Scientific) were reduced with tris(2-carboxyethyl)phosphine hydrochloride (TCEP), alkylated in the dark with iodoacetamide and applied to S-Traps (mini; Protifi) for tryptic digestion (sequencing grade; Promega) in 50 mM triethylammonium bicarbonate (TEAB). Peptides were eluted from the S-Traps with 0.2% formic acid in 50% aqueous acetonitrile and quantified using Pierce™ Quantitative Fluorometric Peptide Assay (Thermo Scientific). Data-independent acquisition mass spectrometry (DIA-MS) was conducted on a Thermo Scientific Orbitrap Exploris 480 mass spectrometer. Online separation was accomplished with a Vanquish Neo UHPLC system (Thermo Scientific): column, PepSep (Bruker; ReproSil C18, 15 cm × 150 µm, 1.9 µm beads); mobile phase A, 0.5% acetic acid (HAc)/0.005% trifluoroacetic acid (TFA) in water; mobile phase B, 90% acetonitrile/0.5% HAc/0.005% TFA/9.5% water; gradient 3–42% B in 120 min; flow rate, 0.4 µl/min. A pool was made of the three control (DMSO) samples, and 2-µg peptide aliquots were analyzed using gas-phase fractionation and 4-m/z windows (30k resolution for precursor and product ion scans) to create an empirically-corrected DIA chromatogram library[72] by searching against a Prosit-generated predicted spectral library[73] based on the UniProt_Human_ref 9606_20220216 protein sequence database (20,588 sequences; 11,394,277 residues). Experimental samples were blocked by replicate and randomized within each block for sample preparation and analysis. Injections of 2 µg of peptides and a 120-min HPLC gradient were employed. MS data for experimental samples were acquired in the orbitrap using 8-m/z windows (staggered; 30k resolution for precursor and product ion scans) and searched against the chromatogram library.

**Data processing and analysis.** Scaffold DIA (v3.3.1; Proteome Software) was used for all DIA-MS data processing. Cysteine carbamidomethylation was set as a fixed modification; trypsin was specified as the proteolytic enzyme, with one missed cleavage allowed. Statistical analysis was performed using Linear Models for Microarray Data (limma) with empirical Bayes (eBayes) smoothing to the standard errors[71]. Proteins with *p*-value < 0.05 and fold change >1.5 were considered to be significant.

## NanoBRET ternary complex formation assay

Plasmids HaloTag-VHL (Cat. No. CS1679C155) and CMV HiBiT (Cat. No. CS1956B03) were purchased from Promega. HiBiT-BCL-xL and HiBiT-BCL-2 were constructed previously[35]. 293T cells ($8 × 10^5$ cells/well in a 6-well plate) were transfected with Lipofectamine (Life Technologies) and 1 µg HaloTag-VHL, 20 ng HiBiT-BCL-xL and 20 ng LgBiT or 1 µg HaloTag-VHL, 20 ng HiBiT-BCL-2 and 20 ng LgBiT. After 48 h, $2 × 10^4$ transfected cells were seeded into each well of white 96-well tissue culture plates in Gibco™ Opti-MEM™ I Reduced Serum Medium, No Phenol Red (Cat. No. 11-058-021, Fisher) containing 4% FBS with or without HaloTag NanoBRET 618 Ligand (Cat. No. PRN1662, Promega) and incubated overnight at 37 °C, 5% $CO_2$. The following day, proteasome inhibitor MG132 (10 µM) was used to pretreat the cells for 2 h and then serial diluted compounds were added into the medium and plates were incubated at 37 °C, 5% $CO_2$, for 4 h. After treatment, NanoBRET Nano-Glo Substrate (cat. no. N1662, Promega) was added into the medium, and the contents were mixed by shaking the plate for 30 s before measuring donor and acceptor signals on Biotek plate reader. Dual-filtered luminescence was collected with a 450/50 nm bandpass filter (donor, NanoBiT-BCL-XL protein or NanoBiT-BCL-2 protein) and a 610-nm longpass filter (acceptor, HaloTag NanoBRET ligand) using an

integration time of 0.5 s. mBRET ratios were calculated following the NanoBRET™ Nano-Glo® Detection System (Cat. No. N1662, Promega).

## Reporting summary

Further information on research design is available in the Nature Portfolio Reporting Summary linked to this article.

## Data availability

Atomic coordinates and structure factors for VCB/753b/BCL-xL, VCB/753b/BCL-2 and VCB/WH244/BCL-2 have been deposited in the Protein Data Bank (PDB) under accession number 8FY0, 8FY1, and 8FY2, respectively. Proteomics data for DIA-MS has been submitted in MASSIVE database (http://massive.ucsd.edu) with database identifier MSV000094144. Proteomics data for TMT-MS has been submitted in PRIDE database (https://www.ebi.ac.uk/pride/) with the database identifier PXD049976. Structural data used from RCSB for published structures referred to in this work are: 1VCB, 1R2D, and 5JSN. All data generated and analyzed in this study are included in this manuscript. Source data are provided with this paper.

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

## Acknowledgements

The authors thank members of the Zheng, Zhou, and Olsen laboratories for helpful discussions. Research reported in this publication was supported by NIH grants R01 GM115568 and R01 GM128731 and CPRIT Rising Star Award RR200030 (S.K.O.). R01 CA242003, R01 CA241191, and R01 AG063801 (G.Z. and D.Z.). NCI R21 CA286307 and a Mays Cancer Center Early Career Pilot Award from CCSG (NIH P30 CA054174) (D.L.). The X-ray diffraction data were collected on beamline NE-CAT 24-ID-E at the Advanced Photon Source, Argonne National Laboratory. The DIA Mass spectrometry analyses were conducted at the University of Texas Health Science Center at San Antonio Institutional Mass Spectrometry Laboratory, with expert technical assistance of Sammy Pardo and Dana Molleur, supported in part by NIH grant P30 CA54174-23 (S.T. Weintraub, Mays Cancer Center Mass Spectrometry Shared Resource) and NIH grant S10 OD030371-01A1 (S.T. Weintraub) for purchase of the Orbitrap Exploris 480 mass spectrometer. This work is based upon research conducted at the Northeastern Collaborative Access Team beamlines, which are funded by the National Institute of General Medical Sciences from the National Institutes of Health (P30 GM124165). The Eiger 16M detector on 24-ID-E is funded by an NIH-ORIP HEI grant (S10OD021527). This research used resources of the Advanced Photon Source, a U.S. Department of Energy (DOE) Office of Science User Facility operated for the DOE Office of Science by Argonne National Laboratory under Contract No. DE-AC02-06CH11357. This research utilized resources of the Structural Biology Core Facilities, part of the Institutional Research Cores at the University of Texas Health Science Center at San Antonio supported by the Office of the Vice President for Research and the Mays Cancer Center Drug Discovery and Structural Biology Shared Resource (NIH P30 CA054174) and the Center for Innovative Drug Discovery (CPRIT Core Facility Award RP210208). The Rigaku HyPix-6000HE Detector, Universal Goniometer, and VariMax-VHF Optic instrumentation in the Structural Biology Core Facilities are funded by NIH-ORIP SIG Grant S10OD030374. The content of this study

is solely the responsibility of the authors and does not necessarily represent the official views of the NIH.

## Author contributions

Protein purification and crystallization experiments were conducted by D.N. and Z.L. Structural experiments and analyses were conducted by D.N., E.A.R., and S.K.O. D.N. and A.N. conducted in vitro biochemical assays. D.L. conducted cell-based studies. Y.Y. conducted in silico experiments which led to design of WH244. P.Z. and W.H. designed and synthesized 753b and WH244. P.S. and R.H. assisted with experimental design and data interpretation. The figures and manuscript were prepared by D.N., D.L., G.Z., D.Z., and S.K.O., with input from all authors.

## Competing interests

G.Z., R.H., and D.Z. are co-founders of and have equity in Dialectic Therapeutics, which develops BCL-xL/2 PROTACs to treat cancer. Other authors declare no competing interests.
