## [Peer Review File · Nature Communications]

Reviewers' Comments:

Reviewer #1:

Remarks to the Author:

Structure-guided development of a more potent second-generation dual degrader of BCL-2 and BCL-xL

This study by Nayak and Lv et al. reports an interesting structural and mechanistic analysis of two VHL-based PROTAC dual degraders (designated 753b and WH244) of the pro-survival BCL-2 family proteins BCL-XL and BCL-2. The development and degradation activity of the PROTAC 753b has previously been reported (Lv et. al 2021 Nat Comms), however in this manuscript the authors extend the previous analysis by determining crystal structures of VHL/753b/BCL-XL and VHL/753b/BCL-2 ternary complexes. The PROTAC WH244 is a novel and more potent derivative, the development of which is described in this manuscript based on structure-guided linker redesign guided by these crystal structures. To authors additionally develop an in vitro ubiquitination assay (PolyUb chain assay) using the VHL/ElonginC/ElonginB complex as well as relevant BCL-XL/BCL-2 point mutants towards validating the PROTAC ternary complex interfaces observed in these structures. Dose-response and time-course data from cellular degradation experiments of BCL-XL and BCL-2 and cell viability data in a sensitive Jurkat cell line are provided to validate the enhanced activity of WH244 relative to 753b. Control experiments are performed using competitor ligand and proteasome/neddylolation inhibitors to indicate that degradation of BCL-XL and BCL-2 by WH244 occur using an on-target PROTAC mechanism. Additionally, the authors generate knock-in HeLa cell lines incorporating the Hibit tag into the endogenous BCL-XL and BCL-2 loci, which are used for dose-response and time-course experiments to further support the enhanced potency of WH244 relative to 753b. An additional ternary structure is solved for the VHL/WH244/BCL-2 complex, which interestingly displays a very different ternary complex interface from that of VHL/753b/BCL-2 that is more similar to the VHL/753b/BCL-XL structure. It is posited that this may be due to additional rigidity and polarity of the WH244 linker to promote this complex and, via modelling the relevant Ub/E2/E3 complex suggested that the closer lysine proximity in this VHL/WH244/BCL-2 complex may promote more effective ubiquitination and degradation.

Overall, this is a well presented and written study with a logical development. Data is well presented to support the degradation activity of the two BCL-XL and BCL-2 dual degraders. Strategies for rational optimisation of PROTACs, in particular using structure-guided approaches remain of significant interest to the TPD field. However, in my view there are some oversights in this particular structural analysis and insufficient supporting data to elaborate on the mechanistic analysis that limit the conclusions of this study, such that overall, it falls short of advancing the field sufficiently to merit publication in this journal.

1. Structural data. The protein/ligand structure data presented in this manuscript, including data acquisition/processing and PDB refinement/deposition statistics are appropriate and well described. However, some concerns must be noted with respect to the structural interpretation and mutagenesis/PolyUb assay data used to validate observed model interfaces.

Structure interpretation. A significant oversight is the absence in analysis of the possible influence of crystal packing on the observed ternary complex interfaces for each of the complexes. Inspection of each of the provided structural models (8FY0, 8FY1 and 8FY2) suggest some extensive interfaces of crystallographically-related protein copies at the ternary complex interface or directly with the PROTAC. E.g., In the 8FY0 structure it is apparent that there is a close packing of a symmetry related copy of ElonginB against the BCL-XL ligand and BCL-XL itself. The possible influence of crystal packing should be addressed in the manuscript and figures showing the crystal packing (and PISA/buried surface analysis for protein copies packed at the ternary complex interface) included in Supporting Information. Comments in lines 365-369 should be addressed accordingly.

2. Mutagenesis/PolyUb assay. The authors generated relevant point mutants of BCL-XL/BCL-2 generated to validate observed protein/ligand interfaces. In general, the presented PolyUb assay data for these mutants tend to support the described interfaces, however the western blots provided show relatively weak and somewhat uneven blotting (in particular Fig 5d) and this makes

the data less convincing. Evidencing quantitation of this data would be strengthened if all three independent repeats of these PolyUb chain assay blots were shown in the manuscript, alongside a schematic indicating what regions of the blot have been used for densitometry/quantitation. Conclusions regarding relevant interface residues would be better supported if additional data were included using a ternary complex formation assay (e.g., as utilised by the authors in Lv et. al 2021 Nat Comms) using VHL with 753b or WH244 and the relevant BCL-2/BCL-XL interface point mutants, or alternatively quantitative binding/cooperativity studies, as utilised in similar studies (e.g., Gadd et al 2017 Nat Chem Biol). This would further support the conclusions regarding the relevance of these interfaces in solution and also the mechanistic conclusions of the paper (as described below). Demonstrating binary binding data of 753b or WH244 with the BCL-XL or BCL-2 interface mutants is also recommended, to rule out that these mutations do not impact ternary complex formation or ubiquitination due to weakening of the interaction with the pro-survival protein.

3. Mechanistic/potency analysis of WH244. The single chemical iteration to incorporate a more polar and rigid linker to generate WH244 is well described and there is clear data presented to support the conclusion that WH244 displays a moderate enhancement (~ 5 -fold) in potency relative to 753b. However, much of the discussion in the manuscript regarding the possible basis for the enhanced activity of WH244 is highly speculative with little quantitative data to reach clear conclusions on the various possible factors (e.g., additional linker polarity enabling new interface interactions or conferring enhanced solubility, or additional linker rigidity possibly locking favourable ternary complex conformations, coupled with more accessible lysine positioning). The ubiquitination assays alone do not adequately address these possibilities, as they do not uncouple binding affinity/cooperativity/ternary complex stability/conformation/lysine accessibility. As noted earlier, binary binding/cooperativity data using one of the many biophysical techniques described in the literature (e.g., Alpha/TR-FRET/FP/ITC/SPR/BLI) or ternary complex formation assays (e.g., Lv et al 2021 Nat Comms; Gadd et al. 2017 Nat Chem Biol) for 753b and WH244 and BCL-XL/BCL-2 interface mutants would better address the question regarding relative complex stability. Solubility data or permeability data could be obtained for the two compounds to address these aspects.

4. Active E3-ligase model. The possibility of lysine proximity being a dominant contributor towards enhancing activity of WH244 is an intriguing one, but at present the model provided in Fig6d is highly speculative and should be removed without additional supporting data to validate the differences (e.g., whether a ~ 12.5 Å difference in lysine positioning significantly affects degradation activity). In their earlier publication the authors described a model for the 753b ternary complex with VCB and BCL-XL or BCL-2 using PROsettaC modelling (Lv et. al 2021 Nat Comms). They should comment whether the new crystallographically-determined model for this complex differs in any material respects – in particular, regarding the relative positioning Lys17 of BCL2. In this earlier study (Lv et. al 2021 Nat Comms), it was previously shown that 753b can polyubiquitinate WT and K22R BCL-2 but not K17R and K17/K22R BCL-2 in 293T cells, suggesting that at least for 753b, Lys17 would seem to be the more relevant of the two (Lys17 or Lys22, BCL-2), although in the new crystallographic ternary complex model presented in Fig6d Lys17 lies more distant than Lys22 by ~ 12.6 Å. This seems counter to the proposition that for WH244 the enhanced activity of towards BCL2 degradation is due to positioning Lys17 closer by an equivalent distance. Perhaps to support this model ubiquitination assays could be repeated using previously described BCL-2 or BCL-XL lysine point mutants (e.g., Lv et al 2021 Nat Comms) to explore for WH244 whether the most proximal lysine is most important for degradation for either BCL-2 and BCL-XL, or incorporating additional lysine residues in closer proximity to the active E2/E3 to see whether the rate or level of ubiquitination is enhanced. Time-course/dose-response comparison of 753b and WH244 for ubiquitination of BCL-XL or BCL-2 may indicate whether ubiquitination rate is a dominant effect on differential degradation activity. If the proposed mechanism holds true, WH244 should display more rapid ubiquitination of BCL-2 due to the closer proximity of Lysine residues.

Other comments

- Throughout the manuscript, the term "PPI" is used to describe interactions both between proteins but also protein and PROTAC/linker interactions (e.g., line 376 "most of the PPIs were centred on the linker"), which is not technically correct – these references to PPIs that are instead protein/ligand interactions should be amended.

- References 22-25 are not the most appropriate to overview other SMIs targeting BCL-2 family proteins, there might be more wholistic reviews to cite?
- Line 72: ABT199 – would be better to refer to this molecule as venetoclax now that it has FDA approval and instead of reference 19 would be better to use the original reference in which this molecule was described. E.g., Park et al. 2008 J Med Chem (doi: 10.1021/jm800669s)
- Line 66-69, 57-59, 67-69 - citations needed.
- Line 75: description of on-target thrombocytopenia - would be better to mention the molecular basis for this effect, with a relevant citation, such as Mason et al Cell 2007 (doi: 10.1016/j.cell.2007.01.037)
- Line 90: other relevant publications exploring the structural/mechanistic basis of VHL-based PROTACs should be cited. E.g., DOI: 10.1038/nchembio.2329; DOI: 10.1038/s41589-019-0294-6; DOI: 10.1126/scitranslmed.abj1578
- Fig 1b, c – the electron density maps should note what sort of maps these are and how these were derived – e.g., are they 2Fobs-Fc maps contoured at 1σ , or are they Fobs-Fc omit maps contoured at 1σ ? Were these omit maps generated from a simulated anneal refinement of an advanced structure, or from the original model prior to ligand building? If these are not 'true' omit maps, then such omit maps should also be included.
- Line 120: the second half of this statement should be less strong as it is not correct - there are small and growing (certainly not zero) number of PROTAC and molecular glue ternary crystal structures in the PDB (e.g., 5t35, 6ZHC, 7PI4, 8BDT, 6HAX, 7Q2J, 7JTO ...)
- Fig3e should indicate treatment time.
- Line 176: should show sequence alignment over relevant regions in Supplementary.
- Line 255: the comment regarding the "superior antitumour activity" of WH244 relative to prior PROTACs for BCL-XL/BCL-2 is too broad, as this molecule is only profiled in one single tumour cell line (Jurkat cells) – this comment should be amended or removed unless improved activity in additional relevant tumour cell lines is also demonstrated.
- Fig6c – recommend to the authors to label the relevant residue numbers in the sequence alignment (or present a full sequence alignment in the Supplementary) – as not all residues in the BCL-XL sequence are shown. The basis for the residue colouring is also not clear.
- Lines 352-369: In this section the authors note potential advantages of their in vitro ubiquitination assay relative to FP/ITC/alpha assays – however in many respects this is a false dichotomy as fundamentally these assays do not measure the same parameters (E.g. the ubiquitination assay is not informative on its own whether a molecule is active or inactive because it has gained or lost binding to one of the proteins, or due to some other reason such as ternary complex stability or positioning of recipient lysine residues). It seems unnecessary to present it this as a dichotomy in the discussion when the Ubiquitination assay is certainly an interesting approach, and these different methods can provide complimentary information.
- Line 387 - the suggested relevance of PK to the washout experiment is also not clear and needs to be further elaborated.
- Line 423 – should note the residue numbers for Elongin B and ElonginC.
- Line 551 – (2×10^4) – not clear whether this is cells per well, needs units; the cell culture medium and volume also needs to be noted. Need to also note if any normalisation was performed – e.g., CTG normalisation to account for compound toxicity.
- Line 579 - Polyubiquitination assay – the source/supplier of the E1, E2, Cdc34, neddylyated Cul2-Rbx1 etc proteins should be noted – or if produced recombinantly this should be included in the methods.

Reviewer #2:

Remarks to the Author:

Summary:

This paper describes a "structure-based" approach to characterize BCL-xL/BCL-2 dual degrader 753b and to inform the design of the WH244 PROTAC with improved potency. Overall, the manuscript is well written and structured, and features several insightful ternary crystal structures at high resolution. Yet, in the line with surprises in the WH422 ternary complex and the mutagenesis data the "rational design" aspect of the study needs to be revised and toned down.

Major critique:

1. "Structure-based design":

While structural biology helped to characterize PROTAC binding, the design aspect is overstated. Intriguingly, the designed PROTAC WH244 adapts an xL-like conformation despite being bound to BCL-2 which was unexpected, i.e. by definition not designed or predictable.

The following lines are overstated:

- Lines 349-351 – "Altogether this work provides a good example for iterative SBDD pipeline in PROTAC field of studies."

- Lines 370 – "This study bolsters the power of structure-guided design..."

Here are respective parts around linker design:

- Line 231 – "We reasoned that a rigid linker would maintain the overall tertiary arrangement...";

Line 319 – "collectively, these results demonstrate ... rigid linker recapitulates the orientation of BCL-2 as observed for BCL-xL with a flex linker"; Line 282 – distinct conformation

In line with some unexpected behavior in the mutagenesis experiments (see below) it is worth considering that the structure may capture one from an ensemble of states. The authors seem to suggest something along those lines here:

- Line 367-369 – "our structures captured a stable intermediate in the entire repertoire of confs..."

Hence, it is worth revising the discussion away from an emphasis on a structure-based design pipeline towards a discussion of ensembles, flexibility and a discussion that design remains challenging, in the context of recent computational progress

(<https://doi.org/10.1021/jacs.2c09387>, [10.1021/acs.jcim.2c01386](https://doi.org/10.1021/acs.jcim.2c01386)). Suggesting a cryoEM study seems beyond the scope here but may be interesting to capture different states (in negative stain) that indicate different orientations of the neosubstrate.

Other relevant sections to reconsider are:

- Line 288 – "the structure-guided strategy ... was validated"

- Fig 2e-f – it may be worth displaying hydrophobic contacts in a LigPlus type fashion in the SI, otherwise it may raise the question how did the lack of interactions inspire linker optimization (cf. line 224 – "linker was instrumental")

Considering the uncertainty of how the ternary complex responds changes in PROTAC proposing a model of entire complex (Fig. 6d) seems questionable. The main conclusion appears to be that Lys's are 12 and 6 Å closer – is that significant given the error associated with such a model? What does this rationalization mean for consequences of xL vs 2 preference. Instead of suggesting to construct a model for the missing xL complex as well, I suggest removing Fig. 6D and rewriting lines 321 onwards as it muddles an otherwise excellent structural paper.

2. Mutagenesis:

Mutagenesis results seem incohesive considering that a structure was at hand.

- Fig 5d, Line 304 – why do double and triple mutations of R110A (+E114K and +E114K+Q118) show increased Ub over single?

- Fig. 5d – Why does the E114A mutation increase %polyUb?

- Line 169 – Mention explicitly that individual mutations did not affect polyUb, sometimes even increased it compared to wt (Why?); Fig 2d – would you have expected charge swap mutant Q111K to be lower %?

- Do the authors have mutagenesis data to speculate on WH244 + xL, in the absence of a WH244 ternary structure, since the "interface is conserved"

- It may be worth considering a quantitative TMT proteomics analysis to assess off-targets as well as clarify the effect on BCL-2 vs -xL.

3. Dual degrader of BCL-xL vs BCL-2:

- Clarify the logic of aiming for a dual degrader overall.

o ABT199 is the only FDA approved drug targeting BCL-2 selectively, yet limited utility in solid tumors that depend on xL but not 2 for survival

♣ Consider context: The effect of the PROTACS reported here appears stronger in xL, yet the BCL2 ternary complex of WH244 is used to rationalize.

o ABT263 is a dual inhibitor but has severe side effects such as thrombocytopenia

- Designed PROTACS seem to function mainly via xL, yet the structure of BCL2+WH244 is used for rationalization. The authors have certainly tried to get the WH244 + xL complex (see

mutagenesis).

- Further define/ quantify how interfaces differ across the 3 ternary complexes; consider adding more detailed side-by-side analysis especially including buried surface area... to the SI, with mention in the main text.
- Why are there still (differential) residual effects in the HiBit knock-in cell lines? (line 257etc., Fig 4b,c)
- Fig 4b,c why is the BCL2 effect lower for 753b than in xL despite "conserved interface" (explain Fig 4b-e in context of benefit for dual degrader)

Minor critique:

1. Clarifications:

- Line 80-81 – DT2216 has reduced platelet tox but still has contains ADT263 as part of the PROTAC. While VHL expression is reduced in platelets, has it been shown that the ADT263 part of the PROTAC has reduced affinity (i.e. similar logic to observing the Hook effect)? Also, do the authors show explicitly that WH422 does not induce thrombocytopenia?
- Line 125 – Discuss limitations of adding compound for purification. Generally, such compounds are synthesized in-house at low-ish yields. Please present overview of how much is needed in this example; considering its generality and usefulness, give ballpark when this strategy may be recommended/ achievable across other projects.
- Line 176 – what is the sequence conservation of the binding site? What about at the PPI interface?
- Line 220 – summarize differences of 2 vs xL.
- Fig 3 – Explain why 293T cells vs Jurkat cells were used. What are 293T's xL vs 2 dependencies?
- Fig. 5C since E114 is involved in salt bridge with R69 the proposed interaction with ligand will be diminished. Considering that's one of very few interactions in a complex that has a tiny PPI surface, can the authors speculate how a stable complex is maintained? (maybe a LigPlus plot could help here). Why does the E114A mutation increase %polyUb (Fig. 5d)?
- Line 380 – does the amino-moiety within piperazine interact as expected based on the ternary structures?
- Why change to HiBit in Hela cells compared to 293T used in previous paper?
- It would be good to compare results to normal control cell line to assess the therapeutic window.
-

2. References:

- Line 55-57 – consider breaking down references for easier accessibility
- Line 59 – add reference
- Line 91 – add reference

3. Text edits:

- Fig 2b caption – data "were"
- Line 61 – "large number" is vague (and relative) – specify, cite reference or circumvent by "recent surge", while still vague is less misleading
- Line 74 – ABT236 should be ABT263
- Line 83 – "Formed 'a' ternary complex"
- Line 117 – xL mentioned twice -> should be BCL-2
- Line 125 – ; vs ,
- Line 125 – "using 'a' gel filtration column"
- Lines 126 to 128 – Correct sentence structure, incomplete sentence, remove 'that': "As can be seen from the gel filtration profile THAT the ternary complex was separated from individual subunits and/or binary complexes (between VCB and 753b, BCL-XL/BCL-2 and 753b) (supplementary Fig. 1a-d) and also served as a proof-of-concept that it is a tightly bound complex."
- Line 134 – the structures are overall certainly carefully refined but consider toning down "excellent" with "reasonable" (after cross-checking validation report)
- Line 140 – BCL-2753b vs 753BLC-2 in Fig 1b – check throughout for consistency
- Line 146 – "...involves residues form..." – either remove "involves" or "form"

Line 171 – “ well-folded” instead of wellfolded

Line 183 to 186: Unclear phrasing “Interestingly, the location of C-terminal of BCL-2 in the ternary complex WAS REPLACED BY the N-terminal of BCL-xL in the corresponding VCB/753b/BCL-xL ternary structure, when both the structures were kept in similar orientation (Supplementary Fig. 3b)”

Line 186 – of “the” linker

Line 199 – BCL-xL (needs hyphen)

Line 214 – “more potent” compared to what? Just 753b or everything?

Line 215 – the design (insert “of”)

Line 229 – linker (insert “is”)

Line 259 – cell(“s”) (insert “are”)

Line 266 – destroyed -> degraded

Line 283 – 753b (space or hyphen) bound

Line 288 – “...PPIs observed in the...” (missing info after “the”)

Line 290 – “...ternary complex interface” (delete comma after “interface”)

4. Consistency: 3-letter code vs 1 letter code for amino acids

5. Replace colloquialisms:

Line 226 – “keeping in mind”

Line 237 – “To our delight”

Line 287 – “To our satisfaction”

Line 367 – “we are more than tempted to speculate”

Line 397 – “comes to the rescue”

Reviewer #3:

Remarks to the Author:

This paper by Nayak et al builds on the previous work by the team developing BCL-XL/BCL-2 dual targeting PROTAC. In this new study, they first determined structures of previously-developed PROTAC 753b with both BCL-XL and BCL-2 in the full VCB ternary complex. This revealed interesting differences in the orientation of the POI and different networks of interactions with the linker and VHL, which was validated by mutagenesis. These differences inspired the structure-guided design of WH244 which was shown to be more potent than 753b in a range of cell-based (using 293T cells) and biochemical assays. Finally, they solved the ternary structure of WH244 with BCL-2 to determine the structural basis for the increased potency.

In general, this is an excellent paper that will benefit the PROTAC field as it contributes novel and informative new structures that provide new information on the general design principles of PROTACs and how they can be improved or re-engineered. All of the structural and associated mutagenesis and structure-function studies are well-presented and the data appears to be of high quality and supports the overall contention of the paper. My only major issue with the paper is in the biological characterisation of WH244 in terms of its efficacy (ie cell-killing activity) which was limited to a single cell line (Jurkats) and no in vivo data. Although a lot of work was put into the development of WH244, this made the story somewhat unbalanced given that the ultimate goal of developing such compounds is to improve their anti-cancer effects. I believe further investigation in this aspect of the study is warranted due to the fact that the present data in the Jurkat cells show only a modest overall improvement in cell-killing activity (no EC50 is presented but it looks like about 3-fold).

Accordingly, the authors should show cell-killing data (using 753b and ABT-263 as comparators) for a wider range of cell lines and the inclusion of solid cancer cells would be useful. I realise these are often co-dependent on MCL-1 for their survival, but co-treatment with MCL-1 inhibitors is straightforward and should provide robust data on the potency of WH244 relative to other BCL-XL inhibitors/degraders. Extending these studies to combinations with chemotherapy, as authors have done previously, would also be informative. I understand in vivo studies can be time-consuming but the author's previous work has shown they are capable of undertaking such experiments. In this regard, I feel these could be justified as the altered chemical structure of the PROTAC may

have adverse PK properties that in fact reduce the potency of the PROTAC relative to the previous iterations such as 753b and DT226) despite the apparently improved in vitro activity. As such, some further information on how the new chemistry influences in vivo behaviour is important.

Reviewer 1_Major:

1. Structural data. The protein/ligand structure data presented in this manuscript, including data acquisition/processing and PDB refinement/deposition statistics are appropriate and well described. However, some concerns must be noted with respect to the structural interpretation and mutagenesis/PolyUb assay data used to validate observed model interfaces.

Structure interpretation. A significant oversight is the absence in analysis of the possible influence of crystal packing on the observed ternary complex interfaces for each of the complexes. Inspection of each of the provided structural models (8FY0, 8FY1 and 8FY2) suggest some extensive interfaces of crystallographically-related protein copies at the ternary complex interface or directly with the PROTAC. E.g., In the 8FY0 structure it is apparent that there is a close packing of a symmetry related copy of ElonginB against the BCL-XL ligand and BCL-XL itself. The possible influence of crystal packing should be addressed in the manuscript and figures showing the crystal packing (and PISA/buried surface analysis for protein copies packed at the ternary complex interface) included in Supporting Information. Comments in lines 365-369 should be addressed accordingly.

Response: We appreciate the reviewer's comments and suggestions. As requested, we have expanded the discussion section and have included commentary on the potential effects of crystal packing. We also prepared Supplementary Figure 14 to address this, where we show different crystallographically related EloB molecules engaging in contacts with the proteins/PROTAC in the asymmetric unit. We have also included a table listing buried surface areas in this figure.

2. Mutagenesis/PolyUb assay. The authors generated relevant point mutants of BCL-XL/BCL-2 generated to validate observed protein/ligand interfaces. In general, the presented PolyUb assay data for these mutants tend to support the described interfaces, however the western blots provided show relatively weak and somewhat uneven blotting (in particular Fig 5d) and this makes the data less convincing. Evidencing quantitation of this data would be strengthened if all three independent repeats of these PolyUb chain assay blots were shown in the manuscript, alongside a schematic indicating what regions of the blot have been used for densitometry/quantitation. Conclusions regarding relevant interface residues would be better supported if additional data were included using a ternary complex formation assay (e.g., as utilised by the authors in Lv et al 2021 Nat Comms) using VHL with 753b or WH244 and the relevant BCL-2/BCL-XL interface point mutants, or alternatively quantitative binding/cooperativity studies, as utilised in similar studies (e.g., Gadd et al 2017 Nat Chem Biol). This would further support the conclusions regarding the relevance of these interfaces in solution and also the mechanistic conclusions of the paper (as described below). Demonstrating binary binding data of 753b or WH244 with the BCL-XL or BCL-2 interface mutants is also recommended, to rule out that these mutations do not impact ternary complex formation or ubiquitination due to weakening of the interaction with the pro-survival protein.

Response: We greatly appreciate the Reviewer's suggestions. We repeated the western blot experiments at least twice to validate the findings and provide cleaner images. The new representative western blot images can be found in Supplementary Fig. 5 and the quantification of all of the repeats can be found in the Source Data file. Also, in Supplementary Fig. 5 we have highlighted the region which was used for quantification in Figs. 2b, 2e and Fig. 5d as red boxes. All of these new data support our previous conclusion that the residues observed in the protein/ligand interface are crucial for activity. In addition, as suggested by the reviewer, we performed the AlphaLISA assays to assess the effects of mutations on binary complex (Supplementary Fig. 7, 11 and 12) and ternary complex formation (Fig. 2c, f and Fig. 5e). Some of the mutants exhibit slightly different binary K_d values for 753b and WH244 binding in the AlphaLISA assay compared to wild-type (WT) BCL-XL and BCL-2, respectively. Moreover, the results from these assays also showed that although all the mutants could form a ternary complex, they were weaker as compared to WT as evidenced by reduction of Alpha signal.

3. Mechanistic/potency analysis of WH244. The single chemical iteration to incorporate a more polar and rigid linker to generate WH244 is well described and there is clear data presented to support the conclusion that WH244 displays a moderate enhancement (~ 5-fold) in potency relative to 753b. However, much of the discussion in the manuscript regarding the possible basis for the enhanced activity of WH244 is highly speculative with little quantitative data to reach clear conclusions on the various possible factors (e.g.,

additional linker polarity enabling new interface interactions or conferring enhanced solubility, or additional linker rigidity possibly locking favourable ternary complex conformations, coupled with more accessible lysine positioning). The ubiquitination assays alone do not adequately address these possibilities, as they do not uncouple binding affinity/cooperativity/ternary complex stability/conformation/lysine accessibility. As noted earlier, binary binding/cooperativity data using one of the many biophysical techniques described in the literature (e.g., Alpha/TR-FRET/FP/ITC/SPR/BLI) or ternary complex formation assays (e.g., Lv et al 2021 Nat Comms; Gadd et al. 2017 Nat Chem Biol) for 753b and WH244 and BCL-XL/BCL-2 interface mutants would better address the question regarding relative complex stability. Solubility data or permeability data could be obtained for the two compounds to address these aspects.

Response: We thank the reviewer for their insightful comments and suggestions. As suggested, we first performed the AlphaLISA assay to analyze the effects of mutations on binary/ternary formation and found that although all the BCL-2 and BCL-xL mutants could form the ternary complex, they were weaker as compared to WT proteins (please see Fig. 2c, Fig. 2f and Fig. 5e). Moreover, some of the mutants exhibit slightly different binary binding affinities to the PROTACs compared to the WT protein (Supplementary Fig. 7, Supplementary Fig. 11 and Supplementary Fig. 12). These differences in binary binding appear to have no significant effect on their ability to form ternary complex. In addition, the AlphaLISA assay results also reveal that WH244 forms a more stable/tight complex with both BCL-xL/BCL-2 than 753b (Fig. 4e, f). Furthermore, we performed the NanoBRET assay to determine the stability of the ternary complex induced by WH244 and 753b in cells and found that WH244 could induce the formation of the ternary complex with BCL-xL at a much lower concentration than 753b (Fig. 4g), which may be partially responsible for the increased potency of WH244 in degrading BCL-xL. However, WH244 was only moderately better than 753b in inducing the formation of the ternary complex with BCL-2 (Fig. 4h), even though WH244 formed a much tighter binary complex with BCL-2 as compared to 753b (753b K_i : 23.4 nM vs. WH244 K_i : 3.1 nM) (Supplementary Fig. 11). These new data provide some explanation for why WH244 is more potent than 753b. The differences between WH244 and 753b in binary binding may be partially attributed to the structure modifications in WH244 compared to 753b such as the replacement of 6-mer alkyl chain with dimethylpiperazine and replacement of morpholine with bridged morpholine. Based on our structural data we found that these two changes could be responsible for the increased binary affinity for WH244 which may in turn lead to a tighter ternary complex formation. Based on Fig. 6c, we saw that the bridged morpholine was participating in additional electrostatic contacts to BCL-2. Further, as presented in Fig. 6d, we observed that Glu114 on BCL-2 engages in contacts with the amino group of piperazine linker. To further support this observation, we found that the mutant E114A led to an increase in K_i of more than two-fold for the binary interaction (Supplementary Fig. 12). Collectively, we can conclude that modifications in WH244 led to stronger binary complex which in turn led to a more stable ternary complex as compared to 753b.

We agree with the reviewer's comment that the structural modifications of WH244 can also change its physicochemical properties compared to 753b, which may affect its solubility and cell permeability and thus its potency in degradation of the targets. We therefore made extensive efforts to develop assays to measure the cell permeability of these compounds. As shown below, we have encountered significant difficulties to get the assays work. Specifically, we initially performed the cell permeability assay using the NanoBRET™ TE Intracellular E3 Ligase Assay from Promega (<https://www.promega.com/products/protein-detection/protein-degradation-protacs/nanobret-te-intracellular-e3-ligase-assays/?catNum=N2910>). Unfortunately, the assay using the VHL tracer from Promega yielded some unexpected finding (See Figure R1 below). It appears that no detectable levels of our compounds could be measured in live cells by this assay. Even in permeabilized cells the levels of our PROTACs were also minimal at high concentrations. These results suggest that the permeability of our PROTACs is too low to be detectable by this assay and/or they cannot effectively compete with the tracer due to a lower binding affinity to VHL after the VHL ligand is linked to the warhead to generate the PROTACs.

Figure R1. Permeability assay using the Promega NanoBRET™ TE Intracellular E3 Ligase Assay.

To test this further, we made a major effort to synthesize a number of ABT-263-derived tracers with different linker lengths and link-out positions (See Figure R2 below). However, upon addition to the medium, these tracers rapidly underwent a shift in emission wavelength and color (from pink to blue, refer to Figure R3 below), which suggests that these tracers may either lack stability or undergo structural alterations in water/medium. Thus, currently we are unable to assess the permeability of our PROTACs. In terms of solubility, all our PROTACs exhibit a high degree of hydrophobicity, rendering them insoluble at concentrations exceeding 20 µM in medium. However, our PROTACs are very potent, and they can achieve BCL-xL and BCL-2 degradation in 1-100 nM range within 2-6 hours. They are soluble in this concentration range.

Figure R2. Structure of ABT263-based tracers

Figure R3. Changes in emission wavelength and color of ABT263-based tracers in water

4. Active E3-ligase model. The possibility of lysine proximity being a dominant contributor towards enhancing activity of WH244 is an intriguing one, but at present the model provided in Fig6d is highly speculative and should be removed without additional supporting data to validate the differences (e.g., whether a ~ 12.5 Å difference in lysine positioning significantly affects degradation activity). In their earlier publication the authors described a model for the 753b ternary complex with VCB and BCL-XL or BCL-2 using PROsettaC modelling (Lv et. al 2021 Nat Comms). They should comment whether the new crystallographically-determined model for this complex differs in any material respects – in particular, regarding the relative positioning Lys17 of BCL2. In this earlier study (Lv et. al 2021 Nat Comms), it was previously shown that 753b can polyubiquitinate WT and K22R BCL-2 but not K17R and K17/K22R BCL-2 in 293T cells, suggesting that at least for 753b, Lys17 would seem to be the more relevant of the two (Lys17 or Lys22, BCL-2), although in the new crystallographic ternary complex model presented in Fig6d Lys17 lies more distant than Lys22 by ~ 12.6 Å. This seems counter to the proposition that for WH244 the enhanced activity of towards BCL2 degradation is due to positioning Lys17 closer by an equivalent distance. Perhaps to support this model ubiquitination assays could be repeated using previously described BCL-2 or BCL-XL lysine point mutants (e.g., Lv et al 2021 Nat Comms) to explore for WH244 whether the most proximal lysine is most important for degradation for either BCL-2 and BCL-XL, or incorporating additional lysine residues in closer proximity to the active E2/E3 to see whether the rate or level of ubiquitination is enhanced. Time-course/dose-response comparison of 753b and WH244 for ubiquitination of BCL-XL or BCL-2 may indicate whether ubiquitination rate is a dominant effect on differential degradation activity. If the proposed mechanism holds true, WH244 should display more rapid ubiquitination of BCL-2 due to the closer proximity of Lysine residues.

Response: We agree with the reviewer's suggestion and have removed Figure. 6d and related discussion from the revised manuscript. In addition, according to the reviewer's suggestion, we conducted the ubiquitination assay in cells using various lysine mutants of BCL-2 as done in our previous work (Lv et. al 2021 Nat Comms). The results from this study show that K17 is the primary lysine responsible for WH244-induced BCL-2 ubiquitination and degradation as seen with 753b treatment (Fig. 3h). However, K22 ubiquitination may be also partially involved in WH244-induced BCL-2 degradation, which can also contribute to the increased potency of WH244 in degradation of BCL-2 compared to 753b (Fig. 3h). We agree with the reviewer that measuring the ubiquitination rate may be also useful to understand the differences between WH244 and 753b and thus performed the assay using the Promega nanoBRET-based ubiquitination assay (https://www.promega.com/products/protein-detection/protein-degradation-protacs/ubiquitination-assay/?gclid=CjwKCAjwvrOpBhBdEiWAR58-3CRPkCAOOcRLmD6n00_OWf2JIC44IxTcmkxdm-3zUAmeJxqDIQbsXBoCFc0QAvD_BwE&catNum=ND2690). Unfortunately, we could not get the assay work. Finally, we performed the dose-response assay as suggested by the reviewer and present the data from this assay in Figure. 3b.

Reviewer 1_Minor:

-Throughout the manuscript, the term “PPI” is used to describe interactions both between proteins but also protein and PROTAC/linker interactions (e.g., line 376 “most of the PPIs were centred on the linker”), which is not technically correct – these references to PPIs that are instead protein/ligand interactions should be amended.

Response: We agree with the reviewer’s suggestion and only use PPIs for protein-protein interactions but not protein-ligand interaction in the revised manuscript.

- References 22-25 are not the most appropriate to overview other SMIs targeting BCL-2 family proteins, there might be more wholistic reviews to cite?

Response: More relevant references have been added as suggested by the reviewer.

- Line 72: ABT199 – would be better to refer to this molecule as venetoclax now that it has FDA approval and instead of reference 19 would be better to use the original reference in which this molecule was described. E.g., Park et al. 2008 J Med Chem (doi: 10.1021/jm800669s)

Response: ABT199 has been replaced with venetoclax and the original publication has been added to the references in the revised manuscript.

- Line 66-69, 57-59, 67-69 - citations needed.

Response: The suggested citations have been added to the revised manuscript.

- Line 75: description of on-target thrombocytopenia - would be better to mention the molecular basis for this effect, with a relevant citation, such as Mason et al Cell 2007 (doi: 10.1016/j.cell.2007.01.037)

Response: We appreciate the suggestion and have noted this information with the appropriate citation.

- Line 90: other relevant publications exploring the structural/mechanistic basis of VHL-based PROTACs should be cited. E.g., DOI: 10.1038/nchembio.2329; DOI: 10.1038/s41589-019-0294-6; DOI: 10.1126/scitranslmed.abj1578

Response: Thank you for the suggestion. These citations have been added.

- Fig 1b, c – the electron density maps should note what sort of maps these are and how these were derived – e.g., are they 2Fobs-Fc maps contoured at 1σ , or are they Fobs-Fc omit maps contoured at 1σ ? Were these omit maps generated from a simulated anneal refinement of an advanced structure, or from the original model prior to ligand building? If these are not ‘true’ omit maps, then such omit maps should also be included.

Response: These figures were 2Fobs-Fc maps contoured at 1.5σ . It was a typo in the legends. However, we generated composite omit maps contoured at 3σ to replace the old figures. Specifically, in the revised manuscript, Fig. 1b right, Fig. 1c right, and Fig. 5b represent composite omit maps which were generated via polder map in Phenix.

- Line 120: the second half of this statement should be less strong as it is not correct - there are small and growing (certainly not zero) number of PROTAC and molecular glue ternary crystal structures in the PDB (e.g., 5t35, 6ZHC, 7PI4, 8BDT, 6HAX, 7Q2J, 7JTO ...)

Response: We agree with the reviewer’s suggestions and have modified the statement in the revised manuscript

- Fig3e should indicate treatment time.

Response: We pretreated the cells with VHL ligand or ABT263 for 2 hours, followed by an additional 6-hour treatment with WH244. We have added this info in the legend of Fig. 3e.

- Line 176: should show sequence alignment over relevant regions in Supplementary.

Response: We have prepared sequence alignment file for the entire sequence and added to Supplementary Fig. 3a, Supplementary Fig. 8 and Supplementary Fig. 13.

- Line 255: the comment regarding the “superior antitumour activity” of WH244 relative to prior PROTACs for BCL-XL/BCL-2 is too broad, as this molecule is only profiled in one single tumour cell line (Jurkat cells) – this comment should be amended or removed unless improved activity in additional relevant tumour cell lines is also demonstrated.

Response: We also compared the potency of WH244 and 753b along with ABT263 and DT2216 against H146 small cell lung cancer cells that are co-dependent on both BCL-xL and BCL-2 for survival and found that WH244 is the most potent among these BCL-xL and BCL-2 targeting agents as shown in Figure R4 below. Because this data is part of a separate manuscript, we could not present the data in this manuscript.

Figure R4. Comparison of the cytotoxicity of different BCL-xL and BCL-2 targeting agents in H146 SCLC cells in vitro.

- Fig6c – recommend to the authors to label the relevant residue numbers in the sequence alignment (or present a full sequence alignment in the Supplementary) – as not all residues in the BCL-XL sequence are shown. The basis for the residue colouring is also not clear.

Response: As suggested by the reviewer, we have added the relevant info in the legend of the figure. Also, in order to accommodate other new figures, we have moved this sequence alignment info to Supplementary Fig. 13.

- Lines 352-369: In this section the authors note potential advantages of their in vitro ubiquitination assay relative to FP/ITC/alpha assays – however in many respects this is a false dichotomy as fundamentally these assays do not measure the same parameters (E.g. the ubiquitination assay is not informative on its own whether a molecule is active or inactive because it has gained or lost binding to one of the proteins, or due to some other reason such as ternary complex stability or positioning of recipient lysine residues). It seems unnecessary to present it this as a dichotomy in the discussion when the Ubiquitination assay is certainly an interesting approach, and these different methods can provide complimentary information.

Response: We agree with the reviewer’s suggestion and have removed this section and just mentioned that the *in vitro* ubiquitination assay can be complimentary to other assays used in TPD field to gain more insight into the mechanism of action of new PROTAC molecules.

- Line 387 - the suggested relevance of PK to the washout experiment is also not clear and needs to be further elaborated.

Response: We agree with the reviewer’s concern and thus have deleted the line in our revised manuscript.

- Line 423 – should note the residue numbers for Elongin B and ElonginC.

Response: The residue numbers for Elongin B and Elongin C have been added to the revised method section.

- Line 551 – (2×10^4) – not clear whether this is cells per well, needs units; the cell culture medium and volume also needs to be noted. Need to also note if any normalisation was performed – e.g., CTG normalisation to account for compound toxicity.

Response: Hela HiBiT-BCL-xL and Hela HiBiT-BCL-2 cells (2×10^4 /well in 80 ml DMEM) were seeded into white 96-well tissue culture plates and incubated overnight at 37 °C, 5% CO₂. For dose curve testing, on the following day, serial diluted compounds were added into the cell culture and then plates were incubated at 37 °C, 5% CO₂, for 6 h. The information has been added to the Methods section of the revised manuscript. We would also like to note that the survival of Hela cells is independent of both Bcl-xL and Bcl-2, as demonstrated in our previous study (<https://doi.org/10.1038/s41467-021-27210-x>). Therefore, there is no need to perform normalization in our PROTAC treatment experiments.

- Line 579 - Polyubiquitination assay – the source/supplier of the E1, E2, Cdc34, neddylated Cul2-Rbx1 etc proteins should be noted – or if produced recombinantly this should be included in the methods.

Response: With the exception of Cul2/Rbx1, which was produced and purified via insect cells, all other proteins were expressed and purified from bacteria. These details have been added in the Methods section.

Reviewer 2_Major:

1. “Structure-based design”:

While structural biology helped to characterize PROTAC binding, the design aspect is overstated. Intriguingly, the designed PROTAC WH244 adapts an xL-like conformation despite being bound to BCL-2 which was unexpected, i.e. by definition not designed or predictable.

The following lines are overstated:

- Lines 349-351 – “Altogether this work provides a good example for iterative SBDD pipeline in PROTAC field of studies.”

- Lines 370 – “This study bolsters the power of structure-guided design...”

Here are respective parts around linker design:

- Line 231 – “We reasoned that a rigid linker would maintain the overall tertiary arrangement...”; Line 319 – “collectively, these results demonstrate ... rigid linker recapitulates the orientation of BCL-2 as observed for BCL-xL with a flex linker”; Line 282 – distinct conformation

In line with some unexpected behavior in the mutagenesis experiments (see below) it is worth considering that the structure may capture one from an ensemble of states. The authors seem to suggest something along those lines here:

- Line 367-369 – “our structures captured a stable intermediate in the entire repertoire of confs...”

Hence, it is worth revising the discussion away from an emphasis on a structure-based design pipeline towards a discussion of ensembles, flexibility and a discussion that design remains challenging, in the context of recent computational progress (<https://doi.org/10.1021/jacs.2c09387>, 10.1021/acs.jcim.2c01386). Suggesting a cryoEM study seems beyond the scope here but may be interesting to capture different states (in negative stain) that indicate different orientations of the neosubstrate.

Other relevant sections to reconsider are:

- Line 288 – “the structure-guided strategy ... was validated”

Response: We fully agree with the reviewer’s comments. The ternary complex structure provides some clues on how to better design PROTAC and could be very useful to provide explanation why some PROTACs are better than others. The information may be not sufficient to precisely design the PROTACs. Therefore, we have revised our manuscript to avoid overstatements such as those pointed out by the reviewer.

- Fig 2e-f – it may be worth displaying hydrophobic contacts in a LigPlus type fashion in the SI, otherwise it may raise the question how did the lack of interactions inspire linker optimization (cf. line 224 – “linker was instrumental”)

Response: We have revised our manuscript based on Reviewer 1. We designed the new linker based on rational design that the piperazine group is rigid and will introduce some polarity which may encourage additional electrostatic contact with proteins. Also, based on our 753b-induced ternary complex structures we found that the length of 6-mer alkyl link is ideal and of similar length as dimethyl piperazine, which supported our linker design strategy.

-Considering the uncertainty of how the ternary complex responds changes in PROTAC proposing a model of entire complex (Fig. 6d) seems questionable. The main conclusion appears to be that Lys's are 12 and 6 Å closer – is that significant given the error associated with such a model? What does this rationalization mean for consequences of xL vs 2 preference. Instead of suggesting to construct a model for the missing xL complex as well, I suggest removing Fig. 6D and rewriting lines 321 onwards as it muddles an otherwise excellent structural paper.

Response: We thank the reviewer for their constructive suggestions. Accordingly, we have removed Fig.6d and related discussion.

2. Mutagenesis:

Mutagenesis results seem incohesive considering that a structure was at hand.

- Fig 5d, Line 304 – why do double and triple mutations of R110A (+E114K and +E114K+Q118) show increased Ub over single?

Response: We have revised our western blot experiments per previous reviewer suggestions and based on this the double mutant R110A/E114K loses complete activity. However, the triple mutant R110A/E114K/Q118A retained some activity, although its activity is still significantly reduced compared to WT. We speculate that mutating Gln118 to Ala is somehow stabilizing the ternary complex in solution and restoring some activity. Moreover, based on Supplementary Fig. 12i, the triple mutant binds WH244 more strongly as compared to single mutant (R110A).

- Fig. 5d – Why does the E114A mutation increase %polyUb?

Response: We apologize for the confusion. The data in Fig. 5d are presented as percent of polyUb compared to WT BCL-2, which shows that all the mutants including E114A exhibit a marked reduction in polyUb as compared to WT.

- Line 169 – Mention explicitly that individual mutations did not affect polyUb, sometimes even increased it compared to wt (Why?); Fig 2d – would you have expected charge swap mutant Q111K to be lower %?

Response: In the revised manuscript we have mentioned that individual mutants had similar activity as WT. Moreover, we have redone all our western blot and normalized all readings. Now the mutants have activity either equivalent or less than WT. R69 of VHL makes multiple contact with Q111 and S110 in BCL-xL (Fig. 2d). The mutant Q111A would still retain the interaction between R69 (VHL) and S110 (BCL-xL). However, the mutant Q111K would cause electrostatic repulsion against positively charged R69 (VHL), which would be more disruptive. Hence it was expected that Q111K would have lower activity.

- Do the authors have mutagenesis data to speculate on WH244 + xL, in the absence of a WH244 ternary structure, since the “interface is conserved”

Response: No, we do not have any mutagenesis data to speculate about WH244/BCL-xL ternary structure. With a lack of real structure, we did not attempt to try this. As mentioned earlier, we extensively tried to crystalize the ternary complex of BCL-xL/WH244/VCB, but were unable to get any crystals.

- It may be worth considering a quantitative TMT proteomics analysis to assess off-targets as well as clarify the effect on BCL-2 vs -xL.

Response: We thank the reviewer's suggestion and have done this experiment accordingly. The data from this experiment are presented in Supplementary Fig. 10b. While we were able to successfully identify approximately 6000 proteins, it is worth noting that Bcl-xL and Bcl-2 were absent. This may be attributed to the potentially low TMT-labeling efficiency of the peptides derived from both proteins. To confirm their degradation, we utilized Western Blotting with the same samples, and successfully verified the degradation of both Bcl-xL and Bcl-2 proteins induced by 753b and WH244 (Supplementary Fig. 10c).

3. Dual degrader of BCL-xL vs BCL-2:

- Clarify the logic of aiming for a dual degrader overall.

o ABT199 is the only FDA approved drug targeting BCL-2 selectively, yet limited utility in solid tumors that depend on xL but not 2 for survival

♣ Consider context: The effect of the PROTACS reported here appears stronger in xL, yet the BCL2 ternary complex of WH244 is used to rationalize.

o ABT263 is a dual inhibitor but has severe side effects such as thrombocytopenia

Response: We apologize for the lack of clarity regarding the rationale to generate a dual degrader. We have revised our manuscript to make it clearer that the primary reason to generate a dual degrader is that some leukemia and cancer cells are dependent on both BCL-xL and BCL-2 for survival. Inhibition/degradation of either protein is not sufficient to kill the tumor cells with ABT199 (an FDA approved BCL-2 specific inhibitor) or DT2216 (a BCL-XL specific PROTAC developed by our group), whereas inhibition of both proteins with ABT263 causes on-target thrombocytopenia. By generating a dual degrader, we can more efficiently kill tumor cells co-dependent on BCL-xL and BCL-2 while avoiding the on-target toxicity because the degrader relies on the VHL E3 ligase to degrade these proteins in a cell type specific manner as platelets express minimal levels of VHL. We have revised the manuscript to make this point clearer.

- Designed PROTACS seem to function mainly via xL, yet the structure of BCL2+WH244 is used for rationalization. The authors have certainly tried to get the WH244 + xL complex (see mutagenesis).

Response: We have extensively tried to crystallize the ternary complex of VCB, WH244 and BCL-xL. However, after multiple variations we were unable to get crystals for this complex.

- Further define/ quantify how interfaces differ across the 3 ternary complexes; consider adding more detailed side-by-side analysis especially including buried surface area... to the SI, with mention in the main text.

Response: We thank the reviewer for this suggestion. We have added this information in new Supplementary Fig. 14 and mentioned it in our discussion.

- Why are there still (differential) residual effects in the HiBit knock-in cell lines? (line 257etc., Fig 4b,c)

Response: The residual effects may be partially attributable to the short-term (6 hr) treatment and lower concentrations (<100 nM) of the PROTACS used in the assay to avoid any non-specific effects and a better comparison of the effects induced by WH244 and 753b. If we extend the treatment duration (to 16 or 24 hours) or increase the concentration, we can achieve 90% degradation of BCL-xL/2. However, this may compromise our ability to make a meaningful comparison of the degradation potency between WH244 and 753b. Additionally, it's worth considering that even after proteasomal degradation of the HiBiT-tagged proteins, there may still be some residual intact HiBiT, which could bind to LgBiT and produce detectable signals. This could be another contributing factor to the observed residual effects.

- Fig 4b,c why is the BCL2 effect lower for 753b than in xL despite "conserved interface" (explain Fig 4b-e in context of benefit for dual degrader)

Response: As reported in our previous publication (Lv D et al. Nat Comm, 12: 6896, 2021), 753b is a more potent degrader for BCL-xL than for BCL-2 in part because BCL-xL has two solvent exposed lysine residuals accessible for ubiquitination while BCL-2 only has one.

Reviewer 2_Minor critique:

1. Clarifications:

- Line 80-81 – DT2216 has reduced platelet tox but still has contains ABT263 as part of the PROTAC. While VHL expression is reduced in platelets, has it been shown that the ABT263 part of the PROTAC has reduced affinity (i.e. similar logic to observing the Hook effect)? Also, do the authors show explicitly that WH422 does not induce thrombocytopenia?

Response: In our previous publication (Khan S et al. Nat Med, 25: 1938-1947, 2019), we reported that converting ABT263 into DT2216 resulted in about 8-fold reduction in its binding affinity to BCL-xL and BCL-2 and the cytotoxicity of DT2216 against tumor cells is mainly dependent on VHL-mediated BCL-xL ubiquitination and degradation because the negative control compound of DT2216, which cannot bind VHL and degrade BCL-xL, has no significant cytotoxicity against tumor cells event at 1 uM concentration.

We measured the potency of ABT263, 753b and WH244 against MOLT-4 T-ALL cells, RS4 B-ALL cells and human platelets in vitro (data shown in the table below). We found that WH244 also exhibited a moderate reduction in platelet toxicity but a dramatic increase in toxicity against leukemia cells compared to ABT263, resulting in more than 100-fold increases in therapeutic index.

Compound	IC ₅₀ MOLT-4 (nM)	IC ₅₀ RS4 (nM)	IC ₅₀ Platelets (nM)	Therapeutic Index (IC ₅₀ MOLT4/ IC ₅₀ Platelets)
ABT-263	220	38	310	1
753b	6.7	7	1,700	253
WH244	4.3	3.2	454	106

- Line 125 – Discuss limitations of adding compound for purification. Generally, such compounds are synthesized in-house at lowish yields. Please present overview of how much is needed in this example; considering its generality and usefulness, give ballpark when this strategy may be recommended/ achievable across other projects.

Response: We agree with the Reviewer's point. We were fortunate to have an abundant source of these compounds readily available. However, to extend this strategy to other projects where there might be scarcity of compounds, based on our experience, ~1 mg of PROTAC is a good starting point. For our work, we started with ~1 mg of 753b, mixed with VCB and BCL-2 and performed gel filtration on a 24 mL Superdex 200 Increase column. The amount of ternary complex isolated using this approach was sufficient for a single round of sparse matrix crystallization trials and few refinements of initial crystal hits. We agree with the reviewer that this strategy requires more of the compounds relative to other assays and approaches, but in our hands for this project, crystallization depended on isolation of a homogenous ternary complex. In a scenario where compounds are severely limiting then a similar strategy could be employed using a 2.4 mL Superdex 200 PC 3.2/30 column, which in principle would reduce the amount of compound required by a factor of 10 (e.g. 100 ug).

- Line 176 – what is the sequence conservation of the binding site? What about at the PPI interface?

Response: The PPI interface of BCL-2 and BCL-xL had a sequence similarity of 70.5% and identity of 54.5% respectively.

- Line 220 – summarize differences of 2 vs xL.

Response: Although speculative, there are several potential explanations for the discrepancy in the degrader activity of DT2216 against BCL-xL and BCL-2 between *in vitro* and *in vivo* experiments. One of the major

reasons could be that full-length, membrane associated BCL-xL and BCL-2 were expressed in cells for the nanoBRET assay, whereas recombinant-transmembrane-domain-deleted proteins were used for the AlphaLISA experiment. In addition, some other proteins might interfere with the interaction between BCL-2, DT2216 and VHL in live cells.

- Fig 3 – Explain why 293T cells vs Jurkat cells were used. What are 293T's xL vs 2 dependencies?

Response: We apologize for the mistake because Jurkat cells were used in these assays. The mistake has been corrected in the revised manuscript. As reported in our previous publication (Lv D et al. Nat Comm, 12: 6896, 2021), 293T cells are not sensitive to the inhibition/degradation of BCL-xL and BCL-2.

- Fig. 5C since E114 is involved in salt bridge with R69 the proposed interaction with ligand will be diminished. Considering that's one of very few interactions in a complex that has a tiny PPI surface, can the authors speculate how a stable complex is maintained? (maybe a LigPlus plot could help here). Why does the E114A mutation increase %polyUb (Fig. 5d)?

Response: In addition to E114, R110 and Q118 also makes PPIs with VHL (please see Fig. 5c). Moreover, as shown in Fig. 5d, an R110A mutant completely abolishes polyubiquitination chain formation as compared to WT protein. Collectively, residues E114, R110 and Q118 maintain the stability of ternary complex. We agree that it is somewhat unusual that the E114A mutation somewhat increases %polyUb and efforts to explain this observation would be speculative. However, based on Fig. 5d, E114A retains only 10-12% of the WT activity so the major conclusions drawn from the data are sound.

- Line 380 – does the amino-moiety within piperazine interact as expected based on the ternary structures?

Response: We have modified the text. The amino-moiety of piperazine was indeed making a polar contact with E114 of BCL-2.

- Why change to HiBit in Hela cells compared to 293T used in previous paper?

Response: In our previous publication (Lv D et al. Nat Comm, 12: 6896, 2021), we reported that neither 293T cells nor Hela cells are dependent on BCL-xL and BCL-2 for survival but equally sensitive to 753b-induced degradation of these proteins. Because Hela cells grow faster and are easier to culture than 293T cells, in this study we selected Hela cells for the HiBit assay.

- It would be good to compare results to normal control cell line to assess the therapeutic window.

Response: Normal cells in general are not sensitive to BCL-xL and BCL-2 inhibition/degradation except platelets. For example, the IC50 values of WH244 and 753b against normal human WI-38 fibroblasts are great than 10 μ M. Compared to ABT263, both WH244 and 753b exhibit significant increases in potency against tumor cells but are less toxic to human platelets as shown in the Table above, indicating that they have increased therapeutic window compared to ABT263.

2. References:

Line 55-57 – consider breaking down references for easier accessibility

Response: Thank you, we have followed this suggestion.

Line 59 – add reference

Response: We have added the relevant reference.

Line 91 – add reference

Response: We have added the relevant reference.

3. Text edits:

Fig 2b caption – data “were”

Response: We have modified this.

Line 61 – “large number” is vague (and relative) – specify, cite reference or circumvent by “recent surge”, while still vague is less misleading

Response: Thank you for this suggestion, we have modified as suggested.

Line 74 – ABT236 should be ABT263

Response: We have modified this.

Line 83 – “Formed ‘a’ ternary complex”

Response: We have modified this.

Line 117 – xL mentioned twice -> should be BCL-2

Response: Thank you. We have modified this.

Line 125 – ; vs ,

Response: Thank you, we have modified this.

Line 125 – “using ‘a’ gel filtration column”

Response: We have modified this.

Lines 126 to 128 – Correct sentence structure, incomplete sentence, remove ‘that’: “As can be seen from the gel filtration profile THAT the ternary complex was separated from individual subunits and/or binary complexes (between VCB and 753b, BCL-XL/BCL-2 and 753b) (supplementary Fig. 1a-d) and also served as a proof-of-concept that it is a tightly bound complex.”

Response: We have modified this.

Line 134 – the structures are overall certainly carefully refined but consider toning down “excellent” with “reasonable” (after cross-checking validation report)

Response: We have modified this.

Line 140 – BCL-2753b vs 753BLC-2 in Fig 1b – check throughout for consistency.

Response: Thank you. We have modified this.

Line 146 – “...involves residues form...” – either remove “involves” or “form”

Response: We have modified this.

Line 171 – “ well-folded” instead of wellfolded

Response: We have modified this.

Line 183 to 186: Unclear phrasing “Interestingly, the location of C-terminal of BCL-2 in the ternary complex WAS REPLACED BY the N-terminal of BCL-xL in the corresponding VCB/753b/BCL-xL ternary structure, when both the structures were kept in similar orientation (Supplementary Fig. 3b)”

We thank the reviewer for pointing this out, and have rephrased this statement to make it clearer. What we were essentially trying to convey was that when the VHL subunits of the BCL-xL^{753b} and BCL-2^{753b} structures are superimposed, one can observe that BCL-xL and BCL-2 are rotated nearly 180 degrees relative to each other due to the altered path of the PROTAC. In addition to the 176 degree relative rotation there is also a translation of 13.6 Å. We have replaced the original text with this description which we feel is significantly clearer.

Line 186 – of “the” linker

Response: We have modified this.

Line 199 – BCL-xL (needs hyphen)

Response: We have modified this.

Line 214 – “more potent” compared to what? Just 753b or everything?

Response: Thank you, we have clarified this point in the revised manuscript.

Line 215 – the design (insert “of”)

Response: We have modified this.

Line 229 – linker (insert “is”)

Response: We have modified this.

Line 259 – cell(“s”) (insert “are”)

Response: We have modified this.

Line 266 – destroyed -> degraded

Response: We have changed this.

Line 283 – 753b (space or hyphen) bound

Response: We have modified this.

Line 288 – “...PPIs observed in the....” (missing info after “the”)

Response: We have modified this.

Line 290 – “...ternary complex interface” (delete comma after “interface”)

Response: We have modified this.

4. Consistency: 3-letter code vs 1 letter code for amino acids

Response: We have modified this.

5. Replace colloquialisms:

Line 226 – “keeping in mind”

Response: We have changed this.

Line 237 – “To our delight”

Response: We have modified this.

Line 287 – “To our satisfaction”

Response: We have changed this.

Line 367 – “we are more than tempted to speculate”

Response: We have modified this.

Line 397 – “comes to the rescue”

Response: We have modified this.

Reviewer 3:

1. In general, this is an excellent paper that will benefit the PROTAC field as it contributes novel and informative new structures that provide new information on the general design principles of PROTACs and how they can be improved or re-engineered. All of the structural and associated mutagenesis and structure-function studies are well-presented and the data appears to be of high quality and supports the overall contention of the paper. My only major issue with the paper is in the biological characterisation of WH244 in terms of its efficacy (ie cell-killing activity) which was limited to a single cell line (Jurkats) and no in vivo data. Although a lot of work was put into the development of WH244, this made the story somewhat unbalanced given that the ultimate

goal of developing such compounds is to improve their anti-cancer effects. I believe further investigation in this aspect of the study is warranted due to the fact that the present data in the Jurkat cells show only a modest overall improvement in cell-killing activity (no EC50 is presented but it looks like about 3-fold).

Accordingly, the authors should show cell-killing data (using 753b and ABT-263 as comparators) for a wider range of cell lines and the inclusion of solid cancer cells would be useful. I realise these are often co-dependent on MCL-1 for their survival, but co-treatment with MCL-1 inhibitors is straightforward and should provide robust data on the potency of WH244 relative to other BCL-XL inhibitors/degraders. Extending these studies to combinations with chemotherapy, as authors have done previously, would also be informative. I understand in vivo studies can be time-consuming but the author's previous work has shown they are capable of undertaking such experiments. In this regard, I feel these could be justified as the altered chemical structure of the PROTAC may have adverse PK properties that in fact reduce the potency of the PROTAC relative to the previous iterations such as 753b and DT226) despite the apparently improved in vitro activity. As such, some further information on how the new chemistry influences in vivo behaviour is important.

Response: We thank the reviewer for their positive comments on our work and for the constructive suggestions. The fact that we used only a single cell line in our biological characterization of WH244 efficacy in our original draft of the manuscript was also an excellent point made by Reviewer 2. As noted above, we now also compare the potency of WH244 and 753b along with ABT263 and DT2216 against H146 small cell lung cancer cells that are co-dependent on both BCL-xL and BCL-2 for survival and found that WH244 is the most potent among these BCL-xL and BCL-2 targeting agents as shown in Figure R4 below. Because this data is part of a separate manuscript, we could not present the data in this manuscript.

Figure R4. Comparison of the cytotoxicity of different BCL-xL and BCL-2 targeting agents in H146 SCLC cells in vitro.

Also as noted above, we measured the potency of ABT263, 753b and WH244 against MOLT-4 T-ALL cells, RS4 B-ALL cells, and human platelets in vitro (data shown in the table below). We found that WH244 also exhibited a moderate reduction in platelet toxicity but a dramatic increase in toxicity against leukemia cells compared to ABT263, resulting in more than 100-fold increases in therapeutic index.

Compound	IC ₅₀ MOLT-4 (nM)	IC ₅₀ RS4 (nM)	IC ₅₀ Platelets (nM)	Therapeutic Index (IC ₅₀ MOLT4/ IC ₅₀ Platelets)
ABT-263	220	38	310	1
753b	6.7	7	1,700	253
WH244	4.3	3.2	454	106

Unlike the BCL-xL specific PROTAC DT2216, our dual BCL-xL and BCL-2 PROTACs were not designed to be combined with other chemotherapeutic agents for cancer treatment but were designed as single agent treatment for BCL-xL and BCL-2 co-dependent leukemia and cancer such as small cell lung cancer. This is because neutrophils are primarily dependent on BCL-xL for survival and BCL-2 inhibition by ABT263 caused the on-target and dose-limiting neutropenia in patients when it was combined with chemotherapy. Therefore, in this study, we did not test the combination of 753b/WH244 with chemotherapy.

We are regretful that we could not perform the PK studies as suggested by the reviewer. As pointed out by the reviewer, PK studies are time consuming and also very costly. We performed the PK studies and in vivo evaluation of the antitumor activity for DT2216 because it was optimized and identified as a lead drug candidate for clinical development. Before the PK and in vivo studies, we characterized the ADME and DMPK properties of DT2216 after taking a lot of efforts to develop a proper formulation for parental administration

because DT2216 is not orally bioavailable due to its high hydrophobicity and low solubility. Because 753b and WH244 have not been optimized for clinical development, we have not been able to secure sufficient funding to conduct these time consuming and expensive studies at the present. Upon further optimization, we hope that we can identify a lead dual BCL-xL and BCL-2 PROTAC to perform the ADME and DMPK studies after developing a proper parental administration formulation in the future as suggested by the reviewer.

Reviewers' Comments:

Reviewer #1:

Remarks to the Author:

I congratulate the authors on an interesting and detailed structure/function study characterising dual BCL-XL/BCL-2 degraders.

The authors have made significant changes in the revised manuscript and have sufficiently addressed the major concerns I raised initially – I would recommend the manuscript be accepted. Some remaining minor comments/suggested changes below.

Minor comments.

Line 68. Proapoptotic BAX and BAK are not mentioned, yet BAK is mentioned in line 78 – suggest to include earlier in line 68.

Line 78. “which is hindered due to ABT263” suggest could be changed to “which is disrupted by ABT263”

Line 93. “play an equally crucial role” – suggest “can also play a crucial role”, as may differ in different contexts.

Line 95. “SMIs”. Acronym is not introduced.

Line 236-237. The AlphaLISA data for BCL-XL mutants in Fig2f is very close to that of the WT protein, hard to say there is a significant difference there. The comment in line 236-237 that the mutants “were all forming relatively weaker ternary complexes compared to WT” would be better to acknowledge that the mutants showed only minor reduction/change in the extent of ternary complex formation and so there might be other reasons for the difference in efficiency of ubiquitination.

Line 328 and Supplementary Data - Significant figures in K_i values. The number of significant figures/decimal places varies in the manuscript. Two decimal places in the error/uncertainty value is not really necessary or meaningful. Suggest use of 1 significant figure for the error/uncertainty value and adjust the decimal places in the K_i value to match this.

Line 358. Reference to “Supplementary Fig 2g, 2h”. Check that this is correct.

Line 390. “two-fold reduction in the K_i ” – suggest “two-fold weakening in the K_i ” is less ambiguous of the direction of the change. Additionally, the R110A mutant resulted in a similar change to the K_i , so ascribing this similar 2-fold difference directly to the interaction of E114 with the linker is not necessarily so clear. Suggest this could be improved.

Line 408. “which we also determined the ternary complex structure of with VCB”, suggested rewording to “for which we also determined the ternary complex structure with VCB”

Line 412. “require the use of various biophysical techniques”. As the techniques are a choice by the researchers and not mandatory, perhaps better to reword as “have typically utilised various biophysical techniques”.

Reviewer #2:

Remarks to the Author:

The authors have addressed most of our initial comments.

Here are some of the remaining points worth considering.

Line 241: The title was adjusted to “Rational and structure-guided design of WH244, a more

potent dual degrader of BCL-xL/BCL-2". Adding "Rational and" at the beginning doesn't address our first major revision request to tone down the language around being "structure-guided"; it also adds redundancy. It may be worth revisiting this point across the manuscript to double-check that the tone is cohesive and clear throughout, i.e. we did notice adjustments in other places referring to the use of computational models rather than structural models.

We thank the authors for considering the TMT proteomics experiment. However, the results are puzzling and inconclusive. While they added an explanation into the main text, it may be worth discussing this further or repeating the experiment. The new Western blot using the TMT sample shows degradation as expected but the point of suggesting proteomics was to demonstrate the on-target off-target profile.

The authors give a good explanation about the use of *cmpd* during purification and why this is a great strategy for their case, where *cmpd* is abundant. As the approach is generally interesting for the field (where applicable) and their discussion is worthwhile, we suggest pulling this from the response letter into the main text.

Same goes for the comment in the response letter on the conservation of the PPI interface, that we couldn't find in the main text.

Finally, although it is flagged as resolved there are still inconsistencies in the abbreviation of the amino acid code (line 213 vs line 286 for example) that we had pointed out in the original review.

Line 325: consider rewording "at a very lower" to be more direct and concise.

Reviewer #3:

Remarks to the Author:

The authors have satisfactorily addressed all my questions.

Reviewer #1 (Remarks to the Author):

I congratulate the authors on an interesting and detailed structure/function study characterising dual BCL-XL/BCL-2 degraders.

The authors have made significant changes in the revised manuscript and have sufficiently addressed the major concerns I raised initially – I would recommend the manuscript be accepted. Some remaining minor comments/suggested changes below.

We thank the reviewer for their insightful comments and suggestions which have led to significant strengthening of the manuscript. We have addressed the remaining issues below.

Minor comments.

1. Line 68. Proapoptotic BAX and BAK are not mentioned, yet BAK is mentioned in line 78 – suggest to include earlier in line 68.

We have included the BAX and BAK in the list of pro-apoptotic proteins in Line 68.

2. Line 78. “which is hindered due to ABT263” suggest could be changed to “which is disrupted by ABT263”

We have reworded the text accordingly.

3. Line 93. “play an equally crucial role” – suggest “can also play a crucial role”, as may differ in different contexts.

We have reworded the text accordingly.

4. Line 95. “SMIs”. Acronym is not introduced.

Thank you, we have defined the SMI acronym in Line 73.

5. Line 236-237. The AlphaLISA data for BCL-XL mutants in Fig2f is very close to that of the WT protein, hard to say there is a significant difference there. The comment in line 236-237 that the mutants “were all forming relatively weaker ternary complexes compared to WT” would be better to acknowledge that the mutants showed only minor reduction/change in the extent of ternary complex formation and so there might be other reasons for the difference in efficiency of ubiquitination.

We thank the reviewer for this suggestion and have reworded the text accordingly.

6. Line 328 and Supplementary Data - Significant figures in Ki values. The number of significant figures/decimal places varies in the manuscript. Two decimal places in the error/uncertainty value is not really necessary or meaningful. Suggest use of 1 significant figure for the error/uncertainty value and adjust the decimal places in the Ki value to match this.

Thank you for the suggestion. We have modified all of the Ki and corresponding error values to reflect only one decimal place.

7. Line 358. Reference to “Supplementary Fig 2g, 2h”. Check that this is correct.

Thank you for pointing this out. This was a typo and we have corrected this in the revised manuscript. It should read Supplementary Fig. 4g, 4h.

8. Line 390. “two-fold reduction in the Ki” – suggest “two-fold weakening in the Ki” is less ambiguous of the direction of the change. Additionally, the R110A mutant resulted in a similar change to the Ki, so ascribing this similar 2-fold difference directly to the interaction of E114 with the linker is not necessarily so clear. Suggest this could be improved.

We thank the reviewer for highlighting this and apologize for the confusion. This was a typo as we meant to state that there was a two-fold increase in the K_i . We have corrected the typo and softened the language with respect to the other point about R110A.

9. Line 408. “which we also determined the ternary complex structure of with VCB”, suggested rewording to “for which we also determined the ternary complex structure with VCB”

We have reworded the text.

10. Line 412. “require the use of various biophysical techniques”. As the techniques are a choice by the researchers and not mandatory, perhaps better to reword as “have typically utilised various biophysical techniques”.

We have reworded the text.

Reviewer #2 (Remarks to the Author):

The authors have addressed most of our initial comments.

We greatly appreciate the reviewer’s insightful comments and suggestions which have led to significant strengthening of the manuscript. We have addressed the remaining issues below.

Here are some of the remaining points worth considering.

1. Line 241: The title was adjusted to “Rational and structure-guided design of WH244, a more potent dual degrader of BCL-xL/BCL-2”. Adding “Rational and” at the beginning doesn’t address our first major revision request to tone down the language around being “structure-guided”; it also adds redundancy. It may be worth revisiting this point across the manuscript to double-check that the tone is cohesive and clear throughout, i.e. we did notice adjustments in other places referring to the use of computational models rather than structural models.

We apologize to the reviewer for the confusion. Based on the reviewer’s previous comments, we did tone down the language overall, giving less impression of “rational and structure-guided”. We had also modified the title to “**Development and crystal structures of a more potent second-generation dual degrader of BCL-2 and BCL-xL**” to make it less obvious. However, in this study we did design the new linker based on rational design that the piperazine group is rigid and will introduce some polarity which may encourage additional electrostatic contact with proteins. Also, based on our 753b-induced ternary complex structures we found that the length of 6-mer alkyl link is ideal and of similar length as dimethyl piperazine, which supported our linker design strategy. So, although the linker design of WH244 was more of a rational decision, we did get some clues from our 753b based ternary structures.

2. We thank the authors for considering the TMT proteomics experiment. However, the results are puzzling and inconclusive. While they added an explanation into the main text, it may be worth discussing this further or repeating the experiment. The new Western blot using the TMT sample shows degradation as expected but the point of suggesting proteomics was to demonstrate the on-target off-target profile.

We appreciate the reviewer’s comment and have made efforts to address this issue as described below. Proteomics is a powerful tool for evaluating both the on-target and off-target effects of novel PROTACs. However, in the context of detecting BCL-xL and BCL-2, TMT proteomics method has some challenges. One of them is the low TMT-labeling efficiency for some proteins and peptides, which can lead to inability to detect the proteins by TMT-based proteomics analysis. This limitation has been observed in several other reported studies, in which the authors failed to detect HDAC degradation by PROTACs (PMID: 37572669 and PMID: 34314730). Despite these challenges, our TMT proteomics profiling successfully identified approximately 6,000 proteins. Notably, the results revealed the absence of any other proteins displaying significant down-regulation

following PROTAC treatment, indicating that our BCL-xL and BCL-2 dual PROTACs have limited off-target effects.

With that said, to address the reviewer's valid concern, we made additional effort to characterize the on-target and off-target effects of our BCL-xL and BCL-2 dual PROTACs using a different proteomics analysis technique, i.e. the Data-independent Acquisition (DIA) proteomics approach. This approach can avoid the limitation of TMT-labeling deficiency for some proteins/peptides. Specifically, we conducted DIA proteomics profiling using HeLa cells because these cells are primarily dependent on MCL-1 for survival and thus resistant to apoptosis after BCL-xL and BCL-2 degradation while expressing a higher level of BCL-xL and BCL-2 than Jurkat cells. To our delight, we were able to identify over 7,200 proteins, including both BCL-xL and BCL-2 (**new Supplementary Fig. 10d**), using this method. Notably, both 753b and WH244 demonstrated the ability to degrade BCL-xL and BCL-2, with WH244 exhibiting superior efficacy in the degradation of both proteins (**new Supplementary Fig. 10e**). Additionally, DIA proteomics revealed perturbations in a few other proteins, which suggests the possibility that these proteins may be downstream targets regulated by BCL-xL/2 or could potentially serve as off-targets, warranting further investigation. In summary, this additional analysis not only validated the on-target degradation of BCL-xL and BCL-2 by WH244 and 753b but also contributed to a more comprehensive assessment of potential off-targets of these new PROTACs.

We have added this discussion to the revised manuscript.

3. The authors give a good explanation about the use of compd during purification and why this is a great strategy for their case, where compd is abundant. As the approach is generally interesting for the field (where applicable) and their discussion is worthwhile, we suggest pulling this from the response letter into the main text.

We have added this strategy in our revised manuscript from Line 142-148.

4. Same goes for the comment in the response letter on the conservation of the PPI interface, that we couldn't find in the main text.

We thank the reviewer for this suggestion. We have added this information in Line 394-395.

5. Finally, although it is flagged as resolved there are still inconsistencies in the abbreviation of the amino acid code (line 213 vs line 286 for example) that we had pointed out in the original review.

We apologize for this. We have used one letter code for all amino acids used in the manuscript.

6. Line 325: consider rewording "at a very lower" to be more direct and concise.

We have reworded the text.

Reviewer #3 (Remarks to the Author):

The authors have satisfactorily addressed all my questions.

Thank you.

Reviewers' Comments:

Reviewer #2:

Remarks to the Author:

The authors have addressed all concerns.
Congratulations on this interesting work!